# Lookahead Routing for Large Language Models

**Canbin Huang[1], Tianyuan Shi[1], Yuhua Zhu[1], Ruijun Chen[1], Xiaojun Quan[1,2,*]**
[1]School of Computer Science and Engineering, Sun Yat-sen University, China
[2]Shenzhen Loop Area Institute, China
{huangcb3, shity6, zhuyh53, chenrj8}@mail2.sysu.edu.cn
quanxj3@mail.sysu.edu.cn

## Abstract

Large language model (LLM) routers improve the efficiency of multi-model systems by directing each query to the most appropriate model while leveraging the diverse strengths of heterogeneous LLMs. Most existing approaches frame routing as a classification problem based solely on the input query. While this reduces overhead by avoiding inference across all models, it overlooks valuable information that could be gleaned from potential outputs and fails to capture implicit intent or contextual nuances that often emerge only during response generation. These limitations can result in suboptimal routing decisions, particularly for complex or ambiguous queries that require deeper semantic understanding. To address this challenge, we propose *Lookahead*, a routing framework that "foresees" potential model outputs by predicting their latent representations and uses these predictions to guide model selection, thus enabling more informed routing without full inference. Within this framework, we implement two approaches based on causal and masked language models. Empirical evaluations across seven public benchmarks—spanning instruction following, mathematical reasoning, and code generation—show that Lookahead consistently outperforms existing routing baselines, achieving an average performance gain of 7.7% over the state-of-the-art. Our code is available at `https://github.com/huangcb01/lookahead-routing`.

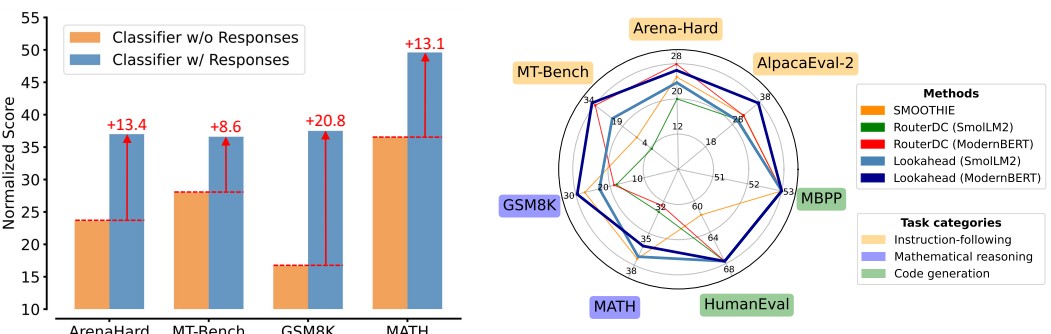

Figure 1: Effect of response-aware routing across benchmarks. **Left:** Including responses improves classifier-based routing performance. **Right:** *Lookahead* outperforms existing routing methods.

## 1 Introduction

Large language models (LLMs) have achieved remarkable success across a wide array of tasks. As different LLMs often exhibit varying strengths, there is growing interest in leveraging multiple LLMs together to build more robust and versatile systems [21]. A straightforward way to combine

---

*   Corresponding author.

multiple LLMs is to query all models in parallel and then select the best response [31, 38]. While this ensemble-based approach can enhance output quality by considering multiple candidate outputs, executing every model for each input query incurs substantial computational cost. To address this, recent work has explored routing-based approaches [9, 17, 30], where a dedicated router selects a single model to handle each query. By directing the input to the most suitable model, these systems aim to retain the benefits of model specialization while significantly reducing inference cost.

Most existing routing methods formulate the task as a classification problem [30, 32, 39], where a router is trained to assign each input query to the most suitable model among a pool of candidate LLMs. However, these approaches typically base the routing decision solely on the input query, without considering how different models would actually respond. As a result, the router lacks access to potentially critical information such as the semantic intent that emerges during generation or the actual quality of the output that each model might produce. This limitation is especially problematic for queries that are ambiguous, underspecified, or require multi-step reasoning, where the true difficulty or requirements of the task only become apparent during response generation. Therefore, this observation raises a fundamental question:

*Should LLM routing depend only on the query, or also consider potential response quality?*

To better understand the limitations of query-only routing, we conduct a preliminary study to examine how access to response content influences the performance of classifier-based routers[2]. We compare two settings: one where the router is trained solely on the input queries, and another where it also has access to the actual responses during training. As shown in Figure 1 (left), normalized scores[3] improve markedly when the router has access to actual responses, which highlights the rich semantic and task-specific information embedded in LLM outputs. Despite these benefits, incorporating response content during inference presents a practical challenge. While actual model responses are not available at test time, generating proxy responses directly from the router is generally infeasible, as the router is typically designed to be lightweight and lacks sufficient generative capability.

To address this limitation, we propose an alternative to generating full responses. Rather than decoding explicit outputs, we train the router to predict the latent representations of potential responses. This approach reduces complexity, as the router only needs to identify key signals linking the input query to likely responses, avoiding the overhead of full-text generation. Based on this insight, we introduce *Lookahead*, a routing framework that allows the router to "foresee" model behavior without performing full decoding. Lookahead is trained to jointly estimate model selection scores and reconstruct the latent features associated with the responses that each candidate LLM would generate. By accessing response-level information in latent space, the router can make more accurate and contextually appropriate routing decisions while maintaining low computational cost.

We instantiate the Lookahead framework in two complementary forms. In the *sequence-level* variant, a small causal language model (CLM) is trained to autoregressively generate a reference response, conditioned on a special model identifier (MID) token. The hidden state at this identifier is extracted and used as a compact representation of the expected response. In the *token-level* variant, a masked language model (MLM) is provided with input sequences in which the entire response is masked using repeated model ID tokens. The model processes this input, and the hidden states are aggregated using attention from the `[CLS]` token to produce a global response representation. Both variants generate response-aware features without requiring explicit output generation during inference. As demonstrated by the empirical results in Figure 1 (right), these features lead to substantial improvements in routing performance, with particularly strong gains from the token-level variant.

In summary, this work reveals the inherent suboptimality of query-only routing strategies and addresses this limitation by introducing Lookahead, a response-aware routing framework. Lookahead predicts the latent representations of model responses without decoding them, and enables more informed routing decisions at reduced computational cost. To support this capability, we propose a dual-task training objective that jointly estimates model selection scores and reconstructs the latent features corresponding to the responses generated by each candidate LLM during training. Empirically, Lookahead consistently outperforms traditional routing baselines across diverse benchmarks, which demonstrates the value of incorporating response-aware signals. Further analysis indicates that Lookahead learns latent representations that simulate the presence of model responses. These representations complement the query and contribute to more accurate routing decisions.

---

[2]Implementation details can be found in Appendix C.2.

[3]Definition can be found in Section 5.1.

## 2   Related work

**LLM ensembling.**   Ensembling [5, 15] is a well-established technique that aims to leverage the diversity and complementary strengths of multiple models to produce more robust and accurate outputs than any single model alone. In the context of LLMs, the motivation for ensembling arises from the observation that different LLMs—even when trained on similar data—often exhibit distinct strengths, biases, and failure modes [21]. Existing ensemble methods can be broadly categorized into static and dynamic strategies, distinguished by whether all models are executed for every input.

Static ensemble methods execute all candidate LLMs for each query and aggregate their outputs post hoc. Within this class, response selection approaches generate one response per model and choose the best. MoRE [31] follows this strategy by training a classifier that scores responses using model expertise, confidence, and inter-response agreement. In contrast, response aggregation methods synthesize a unified output from multiple candidates. LLM-Blender [21] applies pairwise scoring followed by generative fusion; URG [27] combines ranking and rewriting via cross-attention; and LLM-TOPLA [35] improves diversity by pruning near-duplicate candidates. While static ensembles often improve output quality, they incur high computational cost due to full-model invocation.

Dynamic ensemble methods reduce this cost by adaptively selecting which models to run. A common strategy is cascaded inference, where models are ordered by cost and executed sequentially. Execution halts early if a cheaper model's output is deemed sufficient. However, such deferral strategies can be fragile under distribution shift or when model-specific error patterns are not well calibrated [22]. FrugalGPT [7] employs a lightweight verifier to determine whether to halt execution, while Yue et al. [44] validate outputs before escalating to stronger models. Although dynamic ensembles improve efficiency, they often introduce latency and require careful tuning to maintain reliability.

**LLM routing.**   LLM routing [9, 17, 30] aims to optimize efficiency, cost, and performance in model deployment. By dynamically directing inputs to the most suitable model, routing enables lightweight models to handle simpler queries while reserving larger or specialized models for more demanding tasks, thus improving overall resource utilization without compromising output quality. Existing routing methods fall into two main categories: similarity-based and classifier-based approaches.

Similarity-based routing methods rely on the intuition that queries exhibiting similarity to previously observed examples should be directed to LLMs that have historically performed well on such inputs. TO-Router [33] implements this intuition by employing $k$-nearest-neighbor ($k$NN) retrieval over query embeddings and selects the LLM with the highest average performance across the retrieved neighbors. Srivatsa et al. [32] adopted a clustering-based strategy, which uses $k$-means to partition the query space and dispatches queries to the top-performing model within the closest cluster. Eagle [46] extends these approaches by integrating both local (neighborhood-specific) and global Elo-based performance scores to inform routing decisions. SMOOTHIE [17] further eliminates the need for score labels by using response embeddings for similar queries as "voters" to estimate the quality of each LLM through a latent variable graphical model. Despite computationally efficient, these methods often depend on generic query embeddings not optimized for routing-specific objectives.

Classifier-based routing formulates the problem as a supervised text classification task. A pretrained encoder such as BERT [13] is typically fine-tuned to predict which LLMs are likely to perform well, using binary cross-entropy (BCE) [30, 32, 48] or cross-entropy (CE) [28, 39] loss. ZOOTER [26] predicts full reward distributions through a Kullback–Leibler divergence (KLD) objective. RouterDC [9] applies contrastive learning to map queries closer to high-performing LLMs in the representation space. While these methods have demonstrated strong performance, they rely solely on scalar supervision signals and overlook the semantic content of the generated responses.

## 3   Preliminaries

In this section, we begin by formalizing the problem of *LLM routing* and reviewing representative *classifier-based routing* approaches. Our analysis then reveals a fundamental limitation of these methods, which motivates the design of our response-aware *Lookahead* framework.

### 3.1   Problem formulation

Let $X$ denote the space of input queries and $Y$ the space of output sequences. We consider a set of language models $\mathcal{F} = \{f_1, \ldots, f_T\}$, where each model $f_t : X \to Y$ maps an input $x \in X$ to a textual output $y_t = f_t(x)$. A task-specific evaluation function $s : X \times Y \to \mathbb{R}$ assigns a scalar score

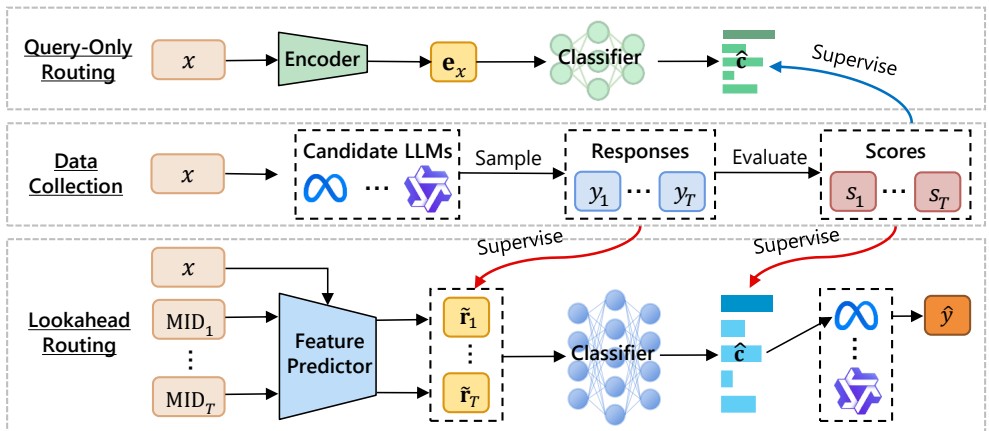

Figure 2: Overview of the *Lookahead* framework. **Middle** (Data Collection): For each input prompt $x$, responses $y_{1:T}$ are sampled from $T$ candidate LLMs. A judge model evaluates these responses to assign quality scores $s_{1:T}$. **Top** (Query-Only Routing): Conventional routers encode $x$ into a query embedding $\mathbf{e}_x$ and select a model based solely on the input. **Bottom** (Lookahead Routing): Given $x$ and model identifiers $\text{MID}_{1:T}$, the Feature Predictor estimates latent response representations $\tilde{\mathbf{r}}_{1:T}$, which are then used by a classifier to predict final quality scores $\hat{c}_{1:T}$ and select the best model.

indicating the quality of each response. For example, $s$ may compute exact-match accuracy for math problems, or output a reward model score in instruction-following tasks.

The goal of LLM routing is to learn a policy $\pi : X \rightarrow \{1, \ldots, T\}$ that selects a model for each input in order to maximize the expected evaluation score:

$$\pi^* = \arg\max_{\pi} \; \mathbb{E}_{x \in X} \left[ s\left( x, f_{\pi(x)}(x) \right) \right]. \tag{1}$$

To train the policy $\pi$, we construct a dataset $\mathcal{D}_{\text{train}} = \{(x^{(i)}, \{(y_t^{(i)}, s_t^{(i)})\}_{t=1}^T)\}_{i=1}^n$, where each $x^{(i)}$ is a sampled input query, $y_t^{(i)} = f_t(x^{(i)})$ is the response from model $f_t$, and $s_t^{(i)} = s(x^{(i)}, y_t^{(i)})$ is the quality score of response $y_t^{(i)}$. This dataset captures both the outputs and corresponding quality assessments across all models, which forms the basis for training the routing policy $\pi$.

### 3.2 Classifier-based routing

Most existing routing approaches cast the problem as a classification task [30, 32, 48]. As illustrated in Figure 2 (top), these methods use an encoder $E : X \rightarrow \mathbb{R}^d$ to map each input query $x \in X$ to a $d$-dimensional embedding $\mathbf{e}_x = E(x)$. A prediction head $C : \mathbb{R}^d \rightarrow \mathbb{R}^T$ then estimates model-specific scores for each candidate model in $\mathcal{F} = \{f_1, \ldots, f_T\}$ and produces a vector: $\hat{\mathbf{c}} = C(\mathbf{e}_x) \in \mathbb{R}^T$, where each component $\hat{c}_t$ approximates the likelihood that model $f_t$ produces a high-quality response. The router then selects the model with the highest predicted score: $\pi(x) = \arg\max_t \hat{c}_t$.

Given supervision targets $\mathbf{c} = [c_1, \ldots, c_T] \in [0, 1]^T$, typically derived from task-specific evaluation scores $\mathbf{s} = [s_1, \cdots, s_T]$ through transformation operations (e.g., binarization or softmax normalization), classifier-based routers are optimized to minimize a pointwise loss:

$$\mathcal{L}_{\text{cls}} = \frac{1}{T} \sum_{t=1}^{T} \ell(\hat{c}_t, \; c_t), \tag{2}$$

where $\ell(\cdot, \cdot)$ denotes a loss function such as cross-entropy [28, 39], binary cross-entropy [30, 32, 48], or Kullback-Leibler divergence [26, 33], depending on the supervision structure.

**Limitation: response agnosticism.** Despite their simplicity and scalability, query-only routers suffer from a fundamental limitation: they ignore the semantic content of the candidate responses $y_t = f_t(x)$. Yet the supervision label $c_t$ inherently depends on the pair $(x, y_t)$, as it reflects the quality of the generated response in context. By conditioning only on the input query $x$, the router is forced to approximate the marginal distribution $p(c_t \mid x)$ without ever observing $y_t$. As a result, it needs to infer how each model in $\mathcal{F}$ behaves based solely on input features. This can lead to overfitting and poor generalization, especially on distributionally shifted or semantically ambiguous queries [30, 48].

# 4 Methodology

We propose *Lookahead*, a response-aware routing framework that bridges the gap between query-only routing and full-response inference. As illustrated in Figure 2, rather than generating complete outputs from all candidate models, Lookahead learns to predict latent representations of each model's response. These predicted features are then used to inform routing decisions. This approach preserves the efficiency of query-only methods while integrating information accessible only via decoding.

## 4.1 The Lookahead framework

As outlined in Section 3.1, the label vector $\mathbf{c}$, derived from the quality scores $\mathbf{s} = [s(x, y_1), \ldots, s(x, y_T)]$, depends on both the input query $x$ and the candidate responses $y_t = f_t(x)$. To capture this dependency without incurring the computational cost of full response generation, the Lookahead framework introduces a predictive module:

$$F : X \times \{1, \ldots, T\} \to \mathbb{R}^k, \quad \tilde{\mathbf{r}}_t = F(x, t),$$

where $\tilde{\mathbf{r}}_t$ denotes a latent representation of the response that model $f_t$ would produce for query $x$.

**Response Modeling.** To ensure that each $\tilde{\mathbf{r}}_t$ captures rich and semantically relevant information, we supervise the predictor $F$ via a response reconstruction objective. Specifically, a decoder $P_D$ is trained to reconstruct the ground-truth response $y_t$ from its predicted latent representation:

$$\mathcal{L}_{\text{resp}} = \frac{1}{T} \sum_{t=1}^{T} \mathcal{L}_{\text{rec}}(x, y_t). \tag{3}$$

Here, $\mathcal{L}_{\text{rec}}$ denotes the reconstruction loss that guides the recovery of $y_t$ from $\tilde{\mathbf{r}}_t$, instantiated as either next-token prediction (in the sequence-level predictor; see Section 4.2) or masked token recovery (in the token-level predictor; see Section 4.3). This auxiliary objective enables the router to learn response-relevant information that complements the query for improved routing decisions.

**Routing head.** Given the query $x$ and the predicted response representations $\{\tilde{\mathbf{r}}_t\}_{t=1}^{T}$, a classifier $C$ estimates the likelihood that each model produces a high-quality response:

$$\hat{\mathbf{c}} = C(x, \tilde{\mathbf{r}}_1, \ldots, \tilde{\mathbf{r}}_T) \in [0, 1]^T.$$

The classifier is trained with binary cross-entropy:

$$\mathcal{L}_{\text{route}} = -\frac{1}{T} \sum_{t=1}^{T} \left[ c_t \log \hat{c}_t + (1 - c_t) \log(1 - \hat{c}_t) \right]. \tag{4}$$

**Joint training objective.** The overall training objective combines routing supervision with auxiliary response modeling:

$$\mathcal{L} = \mathcal{L}_{\text{route}} + \lambda \mathcal{L}_{\text{resp}}, \tag{5}$$

where $\lambda$ is a hyperparameter that balances the importance of response reconstruction.

## 4.2 Sequence-level predictor: causal language model realization

We instantiate the predictor $F$ using a causal language model (CLM) $\text{CLM}_\theta$, which also serves as the decoder $P_D$ (see Figure 3, left). For each candidate model index $t \in [T]$, we concatenate the query with a dedicated model identifier token: $x \parallel \text{MID}_t$.

The CLM is trained under teacher forcing to generate the full response $y_t = (y_{t,1}, \ldots, y_{t,L})$ of length $L$ with the following loss:

$$\mathcal{L}_{\text{rec}} = -\sum_{j=1}^{L} \log P_\theta \left( y_{t,j} \mid x, \text{MID}_t, y_{t,<j} \right). \tag{6}$$

Due to the autoregressive nature of the CLM, the hidden state at the $\text{MID}_t$ position, $\mathbf{h}_t^{\text{MID}}$, encodes the information needed to condition the response generation on both the query and model identity. We adopt this state as the response latent representation: $\tilde{\mathbf{r}}_t = \mathbf{h}_t^{\text{MID}}$. At inference time, this representation is computed in a single forward pass without autoregressive decoding, which enables efficient routing.

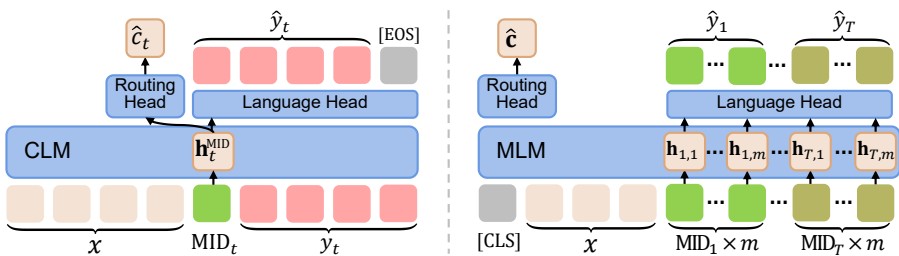

Figure 3: Architectures for response-aware routing in *Lookahead*. **Left**: Sequence-level modeling with a causal language model (CLM), where the hidden state at the MID token encodes the response information. **Right**: Token-level modeling with a masked language model (MLM), where fully masked responses are reconstructed and summarized via attention over MID tokens.

### 4.3 Token-level predictor: masked language model realization

To capture finer-grained semantic distinctions across candidate responses, we also instantiate $F$ as a masked language model (MLM), denoted $\mathrm{MLM}_\phi$ (see Figure 3, right). In contrast to the CLM-based predictor, which models each response separately, the MLM jointly reconstructs all candidate responses in a single forward pass. The input sequence is formed by concatenating the query $x$ with repeated blocks of model identifier tokens:

$$[\texttt{CLS}] \, \| \, x \, \| \, \underbrace{\mathrm{MID}_1, \ldots, \mathrm{MID}_1}_{m \text{ tokens}} \, \| \, \cdots \, \| \, \underbrace{\mathrm{MID}_T, \ldots, \mathrm{MID}_T}_{m \text{ tokens}} .$$

Each block of MID tokens serves as a placeholder for a masked response corresponding to model $f_t$. Let $\mathbf{h}_{t,j}^{\mathrm{MID}} \in \mathbb{R}^d$ denote the hidden state of the $j$-th token in the $\mathrm{MID}_t$ span. These token-level embeddings are stacked to form a matrix representation for the predicted response:

$$\tilde{\mathbf{r}}_t = [\mathbf{h}_{t,1}^{\mathrm{MID}}, \ldots, \mathbf{h}_{t,m}^{\mathrm{MID}}] \in \mathbb{R}^{d \times m} .$$

To produce model selection scores, we prepend a special $[\texttt{CLS}]$ token and extract its hidden state $\mathbf{h}^{\mathrm{CLS}}$, which aggregates information from the full sequence through the attention mechanism. This state is used by a multi-layer perceptron classifier to produce the routing score vector:

$$\hat{\mathbf{c}} = \mathrm{MLP}\left(\mathrm{Attn}\left(\mathbf{h}^{\mathrm{CLS}}, \mathbf{H}_x, \tilde{\mathbf{r}}_1, \ldots, \tilde{\mathbf{r}}_T\right)\right),$$

where $\mathbf{H}_x$ denotes the token-level hidden states corresponding to the input query $x$.

The MLM is trained to reconstruct each fully masked response using the following loss:

$$\mathcal{L}_{\mathrm{rec}} = -\sum_{j=1}^{m} \log P_\phi\left(y_{t,j} \mid x, \mathrm{MID}_t\right). \tag{7}$$

**Curriculum masking.** Unlike conventional MLM pretraining, which masks tokens uniformly at random with a fixed probability $p_0$ (typically 15%), we employ a curriculum masking strategy to better align with the response generation objective. Specifically, we progressively mask from the end of each response and increase the masking ratio linearly to 100% over the first $\alpha$ fraction of training. This approach facilitates a smooth transition from partial to full response masking, which allows the router to learn more robust representations for unseen responses and improve routing performance.

**Design rationale.** The theoretical motivation behind Lookahead stems from the shortcomings of traditional query-only routing, which selects a model based solely on the input query. Lookahead addresses this by introducing a framework that anticipates each model's output through predicted latent representations, which allows the router to "look ahead" without executing full decoding. Because these representations are learned to reflect the underlying semantics of likely outputs, they allow the router to generalize beyond seen data. Both variants of Lookahead—based on causal and masked language models—learn this mapping from input-query context to response-relevant latents. Moreover, the MLM variant employs a progressive masking curriculum, gradually increasing reconstruction difficulty from partial to full responses. This further strengthens the model's ability to form abstract, high-level representations. Together, these components enable Lookahead to achieve more informed and semantically aware routing while maintaining computational efficiency.

# 5 Experiments

## 5.1 Experimental setup

**Candidate LLMs and training data.** We employ five publicly available large language models (LLMs), ranging from 7B to 34B parameters. This selection ensures coverage across a representative spectrum of model capacities and capabilities. Details can be found in Appendix A. To enable the router to effectively leverage the complementary capabilities of the candidate LLMs, we construct a heterogeneous training corpus by aggregating prompts from three publicly available sources spanning diverse domains: (i) *UltraFeedback* [12] provides general-purpose instructions compiled from six widely adopted instruction-tuning datasets. Since these open-ended prompts lack gold-standard references, we assign quality scores using the Skywork-Reward-Gemma-2-27B-v0.2 reward model [24]. (ii) *OpenMathInstruction-2* [36] contains mathematics problems paired with verified solutions. (iii) *Self-Oss-Instruct-SC2* [25] consists of Python programming tasks. Responses are evaluated based on the pass rate of associated unit tests. Details can be found in Appendix C.1.

**Baselines.** We compare Lookahead against six representative routing methods fall into two main categories: (i) *Similarity-based*: $k$NN [33], $k$-means [32], and SMOOTHIE [17]; (ii) *Classifier-based*: multi-label classifier (MLC) [32], ZOOTER [26], and RouterDC [9]. We additionally include three reference methods: (i) *Random Router*, which selects an LLM uniformly. (ii) *Oracle Router*: chooses the LLM with the highest ground-truth score. (iii) *Reward Model Selection*, which selects the highest-scoring response under the reward model Skywork-Reward-Gemma-27B [24].

**Evaluation benchmarks and metrics.** We evaluate routing performance on seven public benchmarks spanning three task types: (i) Instruction-following: AlpacaEval-2 [14], Arena-Hard [23], and MT-Bench [47]. (ii) Mathematics: GSM8K [11] and MATH [19]. (iii) Coding: HumanEval [8] and MBPP [3]. More details of benchmarks and evaluation methods can be found in Appendix B.

We report two evaluation metrics: (i) *Original score* ($\mu_o$), which corresponds to the benchmark's native metric (e.g., *accuracy* or *win rate*) computed over the responses selected by the router; (ii) *Normalized score* ($\mu_n$), which quantifies the proportion of the performance gap between random and oracle routing that a method closes. Specifically, this metric is defined as:

$$\mu_n = \frac{\mu_o - \mu_o^{\text{random}}}{\mu_o^{\text{oracle}} - \mu_o^{\text{random}}} \times 100,$$

where $\mu_o^{\text{oracle}}$ denotes the performance of the Oracle Router. This metric enables consistent evaluation of overall routing effectiveness across benchmarks with heterogeneous scoring scales.

**Implementation.** We implement the CLM-based Lookahead using SmolLM2-135M [2], and the MLM-based variant using ModernBERT-base [41]. For a fair comparison, the classifier-based baselines are reimplemented using the same backbones as Lookahead. The embedding-based baselines utilize the pretrained `all-mpnet-base-v2` model [29]. See Appendix C for further details.

## 5.2 Main results

Table 1 presents both the original and normalized scores of Lookahead, evaluated against strong routing baselines across seven diverse benchmarks. The results highlight two key findings.

**Lookahead consistently outperforms existing routing methods.** Across both architectural settings, Lookahead achieves notable improvements over all baseline approaches. When instantiated with a causal language model (CLM) backbone, it exceeds the performance of the strongest competitor, SMOOTHIE, by 4.5% in average normalized score. In the masked language model (MLM) setting, Lookahead surpasses the best-performing baseline, RouterDC, by a margin of 7.7%. In addition to delivering superior aggregate performance, Lookahead ranks among the top two methods on most of the benchmarks, highlighting its effectiveness and robustness across diverse tasks.

**MLM-based Lookahead provides a decisive advantage in open-ended tasks.** The CLM- and MLM-instantiated versions of Lookahead perform comparably on benchmarks with deterministic evaluation metrics (e.g., mathematics and code). However, on instruction-following tasks—where responses are free-form, reference answers are unavailable, and multiple completions may be equally valid—the MLM variant delivers markedly higher scores. This superiority stems from a core architectural contrast. The CLM variant assigns a likelihood to each candidate in isolation, which prevents reliable, head-to-head comparison across models. By embedding and scoring all completions

Table 1: Performance across seven evaluation benchmarks in both the raw metric ($\mu_o$) and its normalized form ($\mu_n$). The highest $\mu_n$ in each column is shown in **bold**, and the second-highest is underlined. Normalized scores reflect the percentage of performance gain over random routing.

| Method | AlpacaEval-2 | | Arena-Hard | | MT-Bench | | GSM8K | | MATH | | HumanEval | | MBPP | | Avg. |
|---|---|---|---|---|---|---|---|---|---|---|---|---|---|---|---|
| | $\mu_o$ | $\mu_n$ | $\mu_o$ | $\mu_n$ | $\mu_o$ | $\mu_n$ | $\mu_o$ | $\mu_n$ | $\mu_o$ | $\mu_n$ | $\mu_o$ | $\mu_n$ | $\mu_o$ | $\mu_n$ | $\mu_n$ |
| *Candidate LLMs* | | | | | | | | | | | | | | | |
| Yi-1.5-34B-Chat | 37.6 | 28.9 | 44.1 | 26.1 | 7.81 | 9.9 | 88.4 | 13.8 | 51.9 | -4.0 | 69.5 | -17.7 | 70.8 | -5.2 | 7.4 |
| InternLM-2.5-20B-Chat | 37.1 | 27.4 | 28.3 | -9.2 | 7.48 | -20.4 | 89.8 | 27.4 | 62.2 | 37.4 | 72.6 | -3.0 | 68.9 | -14.7 | 6.4 |
| Phi-3-Medium-4K-Instruct | 29.8 | 1.4 | 33.3 | 2.1 | 7.99 | 26.5 | 89.1 | 20.2 | 44.5 | -33.8 | 74.4 | 5.9 | 68.8 | -14.7 | 1.1 |
| Llama-3.1-8B-Instruct | 24.6 | -17.1 | 21.2 | -25.1 | 7.69 | -1.1 | 81.6 | -50.2 | 44.7 | -33.1 | 62.2 | -52.9 | 68.1 | -18.4 | -28.3 |
| Qwen2.5-Coder-7B-Instruct | 18.0 | -40.5 | 35.1 | 6.1 | 7.54 | -14.9 | 85.8 | -11.3 | 61.2 | 33.5 | 87.2 | 67.7 | 82.9 | 53.0 | 13.4 |
| *Reference Methods* | | | | | | | | | | | | | | | |
| Random Router | 29.4 | 0.0 | 32.4 | 0.0 | 7.70 | 0.0 | 86.9 | 0.0 | 52.9 | 0.0 | 73.2 | 0.0 | 71.9 | 0.0 | 0.0 |
| Oracle Router | 57.6 | 100 | 77.1 | 100 | 8.79 | 100 | 97.5 | 100 | 77.8 | 100 | 93.9 | 100 | 92.6 | 100 | 100 |
| Reward Model Select | 41.6 | 43.4 | 55.7 | 52.1 | 8.28 | 53.1 | 93.0 | 57.7 | 67.1 | 56.9 | 83.5 | 50.0 | 77.8 | 28.6 | 48.8 |
| *Routing Methods (Pretrained Embeddings)* | | | | | | | | | | | | | | | |
| kNN [33] | 38.5 | 32.2 | 39.9 | 16.7 | 7.72 | 1.7 | **89.8** | **27.4** | **62.3** | **37.6** | 76.2 | 14.7 | 73.2 | 6.0 | 19.5 |
| k-means [32] | 38.9 | 33.8 | 38.9 | 14.6 | 7.81 | 9.9 | **89.8** | **27.4** | 62.2 | 37.3 | 79.9 | 32.4 | 78.2 | 30.4 | 26.6 |
| SMOOTHIE [17] | 38.5 | 32.1 | 43.6 | 25.0 | 7.83 | 11.3 | **89.8** | **27.4** | 62.2 | 37.4 | 86.0 | 61.8 | **82.9** | **53.0** | 35.4 |
| *Routing Methods (CLM-based, SmolLM2)* | | | | | | | | | | | | | | | |
| MLC [32] | 39.4 | 35.4 | 41.9 | 21.3 | 7.78 | 7.3 | 88.5 | 14.5 | 62.2 | 37.2 | 85.4 | 58.8 | 73.5 | 7.9 | 26.1 |
| ZOOTER [26] | 37.2 | 27.6 | 42.3 | 22.2 | 7.79 | 8.4 | 88.9 | 18.8 | 61.9 | 36.2 | 86.6 | 64.7 | 77.0 | 64.7 | 29.0 |
| RouterDC [9] | 37.9 | 30.1 | 41.3 | 20.0 | 7.74 | 3.3 | 88.9 | 18.1 | 61.1 | 33.0 | **87.2** | **67.7** | **82.9** | **53.0** | 32.2 |
| Lookahead (ours) | 37.8 | 29.8 | 43.0 | 23.7 | 7.97 | 24.5 | 89.4 | 23.1 | 62.2 | 37.2 | **87.2** | **67.7** | **82.9** | **53.0** | 37.0 |
| *Routing Methods (MLM-based, ModernBERT)* | | | | | | | | | | | | | | | |
| MLC [32] | 38.5 | 32.4 | 43.0 | 23.6 | 8.01 | 28.0 | 88.7 | 16.7 | 62.0 | 36.5 | 83.5 | 50.0 | 82.5 | 51.1 | 34.0 |
| ZOOTER [26] | 37.0 | 26.8 | 41.2 | 19.7 | 7.78 | 7.3 | 89.3 | 22.4 | 62.0 | 36.6 | 86.0 | 61.8 | 81.7 | 47.4 | 31.7 |
| RouterDC [9] | 38.4 | 32.0 | **44.9** | **27.9** | 8.07 | 33.7 | 88.9 | 18.8 | 61.0 | 32.4 | **87.2** | **67.7** | **82.9** | **53.0** | 37.9 |
| Lookahead (ours) | **40.0** | **37.5** | 44.3 | 26.5 | **8.09** | **35.4** | **90.1** | **29.6** | 61.9 | 36.2 | **87.2** | **67.7** | **82.9** | **53.0** | **40.8** |

Figure 4: Results of ablation studies for the Lookahead framework. (**a**) Performance drops when response modeling (RM) is removed. (**b**) Comparison of curriculum masking (CM) strategies in the MLM-based predictor. (**c**) Effectiveness of the joint response prediction.

jointly within a shared semantic space, the MLM variant can make fine-grained, context-aware distinctions among outputs, a capability that proves critical for routing under open-ended conditions.

## 5.3 Ablation studies

**Impact of response modeling.** Figure 4 (a) presents the results of ablation studies assessing the contribution of the response modeling objective. Disabling this component leads to significant performance degradation, with absolute drops of 6.2 and 6.8 points for the CLM- and MLM-based variants, respectively. These declines highlight the critical role of response representations in providing meaningful contextual signals for routing decisions. The consistent impact across both architectural paradigms underscores the generality and robustness of the Lookahead framework.

**Impact of curriculum masking.** Figure 4 (b) investigates the effect of curriculum masking within the MLM-based implementation. Removing this mechanism results in a 4.7 points performance drop, indicating that predicting full response sequences is considerably more difficult for models pre-trained on span-level objectives. To better understand the design choices, we compare three

curriculum strategies: (i) End-masking (default): reveals tokens progressively from end to start, (ii) Start-masking: reveals tokens from start to end, (iii) Random masking: reveals random spans of increasing length. While both start-masking and random masking reduce the learning difficulty and yield moderate gains over the no-curriculum baseline, they are less effective than end-masking. We attribute this effectiveness to its alignment with the task objective, as both involve predicting response continuations from prefixes, which leads to more coherent and task-consistent representations.

**Impact of the joint response prediction.** Figure 4 (c) compares joint versus separate prediction of candidate responses within the MLM architecture. Joint prediction, which embeds all responses into a shared latent space, markedly outperforms per-model prediction, especially on instruction-following (IF) benchmarks. Unlike math or code tasks, open-ended tasks lack objective ground truth, so effective routing hinges on preference-driven distinctions across responses. Shared-space encoding enables the attention mechanism to capture these fine-grained semantic contrasts, whereas separate prediction deprives the router of the comparative context required for reliable quality assessment.

### 5.4 Analysis

We analyze the behavior of Lookahead with a causal language model (CLM) backbone to better understand how response modeling affects training efficiency and representation quality. Empirically, both the CLM- and MLM-based variants exhibit similar trends; thus, for clarity and brevity, we focus on the CLM-based implementation here and report MLM results in Appendix D.

**Response modeling improves training efficiency.** Figure 5 (a) quantifies the effect of the response-modeling objective on sample efficiency. With only ∼16% of the training data, the *w/ RM* model achieves the same performance that the *w/o RM* baseline requires the full dataset to reach, resulting in a 6.3× gain in data efficiency. Even at full scale, the *w/ RM* variant delivers a 1.2× performance advantage. These results suggest that auxiliary response reconstruction enables Lookahead to learn more informative latent representations in low-data regimes, leading to more accurate routing with significantly fewer supervision signals.

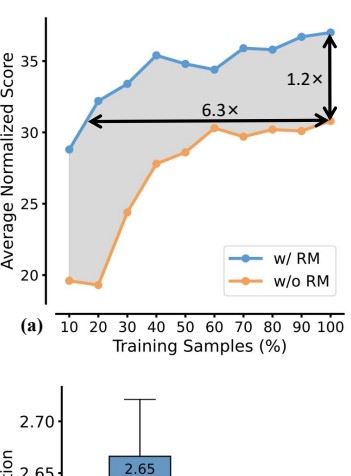

**Response embeddings capture richer semantics than query-only features.** To verify that Lookahead indeed learns response-aware representations, we compare three routing models: (i) *w/ RM*, our full method that incorporates the response-modeling objective; (ii) *w/o RM*, a baseline that relies solely on query features; and (iii) *w/ actual response*, an oracle that selects routes using the true candidate responses. We measure the mutual information (MI) between each learned model's hidden states and those of the oracle with MINE[4], repeating the estimation 50 times with independent seeds to control variance. Figure 5 (b) shows the resulting MI distributions. The *w/ RM* variant achieves a substantially higher MI than the *w/o RM* baseline, as evidenced by non-overlapping interquartile ranges and medians. This confirms that the response-modeling objective encourages $\tilde{\mathbf{r}}_t$ to capture richer semantic information from the response space, which enables Lookahead to bridge the gap between purely query-based routing and full-response inference.

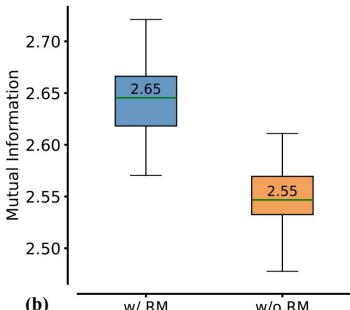

Figure 5: (a) Training efficiency and (b) mutual-information analysis for CLM-based Lookahead.

### 6 Conclusion

We introduced *Lookahead*, a routing framework that enhances model selection in multi-LLM systems by predicting latent representations of potential responses rather than relying solely on input queries. This enables more informed and context-aware routing decisions while avoiding the computational cost of full-text generation. By implementing both causal and masked language model variants, Lookahead demonstrates effectiveness across seven diverse benchmarks, consistently outperforming state-of-the-art routing baselines—with especially strong gains on open-ended tasks where nuanced semantic differences are decisive. These results underscore the value of response-aware latent features and highlight the potential of incorporating lightweight generative foresight into LLM routing.

## Acknowledgments and Disclosure of Funding

This work was supported by the National Natural Science Foundation of China (No. 62176270) and the Guangdong Basic and Applied Basic Research Foundation (No. 2023A1515012832).

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

## A Details of open-source models

Table 2 lists the Hugging Face repository identifiers for candidate LLMs, backbone models of routers, and the reward model employed to evaluate response quality during training set construction.

Table 2: Details of open-source models in our experiments.

| Model | Parameters | Hugging Face ID |
|---|---|---|
| *Candidate LLMs* | | |
| Yi-1.5-34B-Chat [43] | 34.4B | 01-ai/Yi-1.5-34B-Chat |
| Internlm2.5-20B-Chat [6] | 19.9B | internlm/internlm2_5-20b-chat |
| Phi-3-medium-4k-instruct [1] | 14.0B | microsoft/Phi-3-medium-4k-instruct |
| Llama-3.1-8B-Instruct [16] | 8.0B | meta-llama/Llama-3.1-8B-Instruct |
| Qwen2.5-Coder-7B-Instruct [20] | 7.6B | Qwen/Qwen2.5-Coder-7B-Instruct |
| *Router Backbones* | | |
| SmolLM2-135M [2] | 135M | HuggingFaceTB/SmolLM2-135M |
| ModernBERT-base [41] | 149M | answerdotai/ModernBERT-base |
| *Reward Model* | | |
| Skywork-Reward-Gemma-2-27B-v0.2 [24] | 27.2B | Skywork/Skywork-Reward-Gemma-2-27B-v0.2 |

## B Details of evaluation benchmarks and original metrics

- **AlpacaEval-2** [14] contains 805 instructions from five different datasets. In this benchmark, GPT-4-Preview-1106 serves both as the baseline model and the judging model to calculate length-controlled win rate [14] as the metric.

- **Arena-Hard** [23] is a challenging evaluation benchmark whose results closely align with human preference rankings from Chatbot Arena [10]. The benchmark covers 250 high-quality topic clusters, including 500 well-defined technical problem-solving queries. We report the model's win rate relative to GPT-4-0314, using GPT-4-Preview-1106 as the judging model.

- **MT-Bench** [47] consists of 80 multi-turn conversations, totaling 160 questions. Each response is scored by GPT-4 on a scale from 1 to 10, and the average score per conversation is calculated. Unlike the official setup, we follow recent studies [37, 40, 42], using GPT-4-0125-Preview as both the judging model and the baseline model. Following ZOOTER [26], we only route with the first-turn query but evaluate in the multi-turn manner.

- **GSM8K** [11] consists of elementary-level mathematical problems designed to evaluate a model's mathematical reasoning capabilities. The exact match accuracy is calculated as the metric.

- **MATH** [19] contains various mathematical problems ranging from middle school to high school competition levels, comprehensively evaluating the model's mathematical capabilities in areas such as algebra, calculus, number theory, and probability, with accuracy serving as the metric.

- **HumanEval** [8] evaluates a model's code-writing abilities by providing function signatures and docstrings and requiring the model to generate corresponding Python function bodies. We calculate pass@1 as the metric.

- **MBPP** [3] consists of a series of programming problems aimed at testing the model's ability to generate Python code snippets based on natural language descriptions. The metric is pass@1.

## C Implementation details

### C.1 Training data construction

For each query, we sample responses from candidate LLMs with a temperature of 0.8 and top-$p$ of 0.95. For open-ended instructions, we assign quality scores using the Skywork-Reward-Gemma-2-27B-v0.2 reward model [24]. For mathematical problems, we calculate the accuracy by rule-based matching with verified solutions. To ensure diversity in model performance, we exclude prompts on which all LLMs produce identical correctness outcomes. Additionally, we enrich the binary correctness signal with a normalized reward model score to account for cases where an LLM may arrive at the correct

answer via incorrect reasoning [45]. For code generation tasks, responses are evaluated based on the pass rate of associated unit tests, and samples with all LLMs achieving the same score are filtered out. All scores are min–max normalized within each dataset and converted to binary classification labels by comparing with an empirically set threshold of 0.8.

In Table 3, we provide the datasets from which queries are sampled to construct the training set along with the percentage of highest-scoring responses per candidate LLM.

Table 3: Details of datasets used in training and evaluation. The top block shows the number of samples; the bottom block reports the percentage of highest-scoring responses per model. The sum of the percentages for each dataset may exceed 100%, because for some queries, multiple candidate responses can simultaneously achieve the highest score.

| Item | UltraFeedback | OpenMathInstruction-2 | Self-Oss-Instruct-SC2 | Overall |
|---|---|---|---|---|
| *Sample Counts* | | | | |
| Train | 43,757 | 12,189 | 3,335 | 59,281 |
| Validation | 4,862 | 1,354 | 1,000 | 7,216 |
| *Percentage of Highest-scoring Responses* | | | | |
| Yi-1.5-34B-Chat | 29.51 | 14.78 | 27.20 | 26.36 |
| Internlm2.5-20B-Chat | 34.61 | 37.21 | 41.22 | 35.57 |
| Phi-3-medium-4k-instruct | 11.95 | 9.98 | 39.31 | 13.33 |
| Llama-3.1-8B-Instruct | 15.45 | 7.52 | 37.19 | 15.25 |
| Qwen2.5-Coder-7B-Instruct | 10.58 | 30.74 | 47.84 | 17.11 |

## C.2 Preliminary study implementation

The preliminary study in Section 1 compares two settings: (i) The router receives only the input query and generates predictive scores indicating each LLM's potential performance. This paradigm employs a BERT-based multi-label classifier architecture described in Section 3.2. (ii) The router processes both the query and candidate responses to identify optimal answer selections. This approach implements a BERT-based binary classification framework that evaluates concatenated query-response pairs, predicting whether a given response constitutes an appropriate answer to the query. Both models are optimized using binary cross-entropy loss under identical hyperparameter configurations to those employed in the MLM-based Lookahead.

## C.3 Baseline implementation

- **$k$NN** [33] retrieves the $k$ most similar training examples based on query embedding similarity and routes to the LLM with the highest average score on these neighbors.

- **$k$-means** [32] clusters training queries into $k$ groups and routes each test query to the LLM that performs best on its nearest cluster.

- **SMOOTHIE** [17] estimates quality scores using a latent variable graphical model constructed from response embeddings of similar queries.

- **MLC** [32] employs a multi-label classifier trained with binary cross-entropy loss to identify all LLMs likely to perform well on a given query.

- **ZOOTER** [26] predicts the full response score distribution for each LLM using a KL-divergence objective, enabling probabilistic selection.

- **RouterDC** [9] applies contrastive learning to align query embeddings with well-performing LLMs and distance them from poor performers.

## C.4 Hyperparameter tuning

We perform a grid search on the validation set to find the optimal hyperparameters for our proposed Lookahead and baselines. For CLM-based Lookahead, we finally set $\lambda$ to 0.5. For the MLM-based varient, we set $\lambda$ to 0.2, $m$ to 64, and $\alpha$ to 0.4. For $k$NN [33] and $k$-means [32], we set $k$ to 100. For SMOOTHIE [17], we set $n_0$ to 1000. For ZOOTER [26] and RouterDC [9], hyperparameters in the original papers is find to be optimal and adopted.

## C.5 Training details

We conducted routing experiments with a batch size of 64 and a maximum length of 2048 tokens on a single 24GB NVIDIA RTX 3090 GPU. The training was performed on 2 and 4 epochs for CLM- and MLM-based implementations, respectively. A cosine learning rate schedule and the AdamW optimizer are employed with a learning rate of 5e-5. We save checkpoints every 100 steps and select the best one based on validation set performance.

## C.6 Implementation of mutual information estimation

We utilize Mutual Information Neural Estimation (MINE) [4] to quantify the response information captured in Lookahead's hidden states. This approach is preferred over traditional non-parametric methods, which struggle with high-dimensional latent spaces. The MINE estimator is implemented as a multi-layer perceptron (MLP) comprising four fully connected layers, each with 1024 hidden units and ReLU activation functions. Training is conducted using the AdamW optimizer for 100 epochs, with a batch size of 512 and a learning rate of 1e-4. A linearly decaying learning rate scheduler is applied, incorporating a warm-up phase comprising 10% of the total training steps.

# D    MLM-based results for Section 5.4

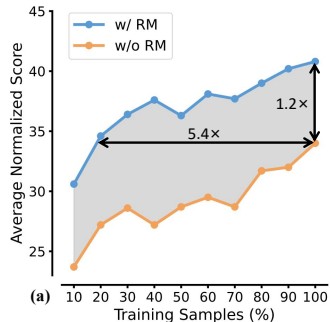 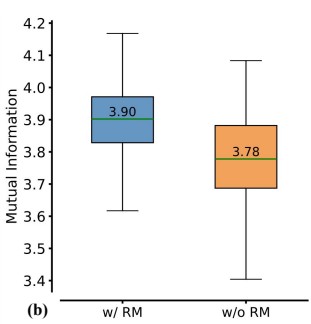

Figure 6: (**a**) Sample efficiency of the MLM-based model with and without response modeling (RM). (**b**) Mutual information between the hidden states of the MLM-based Lookahead and an oracle classifier with access to full responses.

**Training efficiency.**    As shown in Figure 6 (a), the MLM-based Lookahead shows similar advantages to the CLM-based variant in improving sample efficiency. Specifically, with only ∼18.5% of the training data, the *w/ RM* model achieves the same performance that the *w/o RM* baseline requires the full dataset to reach. Even at full scale, the *w/ RM* variant delivers a 1.2× performance advantage. These results demonstrate the remarkable effectiveness in improving training efficiency.

**Mutual-information analysis.**    We use MINE [4] to measure the semantic information Lookahead learned with the same experimental setup described in Section 5.4. As shown in Figure 6 (b), the *w/ RM* variant achieves a substantially higher MI than the *w/o RM* baseline, as evidenced by non-overlapping interquartile ranges and medians. This confirms that the response-modeling objective drives the latent vectors $\tilde{\mathbf{r}}_t$ to capture richer semantic information from the response space.

# E    Effect of hyperparameters

## E.1    Effects of $\lambda$

We conduct an experiment to study the effect of $\lambda$ in Eq. 5 w.r.t. the average normalized score. As shown in Figure 7 (a), the overall performance of Lookahead is insensitive to a wide range of $\lambda \in [0.1, 0.6]$ for both implementations, making it easy to choose the value of $\lambda$ in practice.

## E.2    Effects of $m$

The results in Figure 7 (b) demonstrate the impact of varying the number of repeated model ID tokens $m$ on routing performance. Initially, increasing $m$ improves the average normalized score as the router models longer response sequences, enabling it to capture more comprehensive semantic

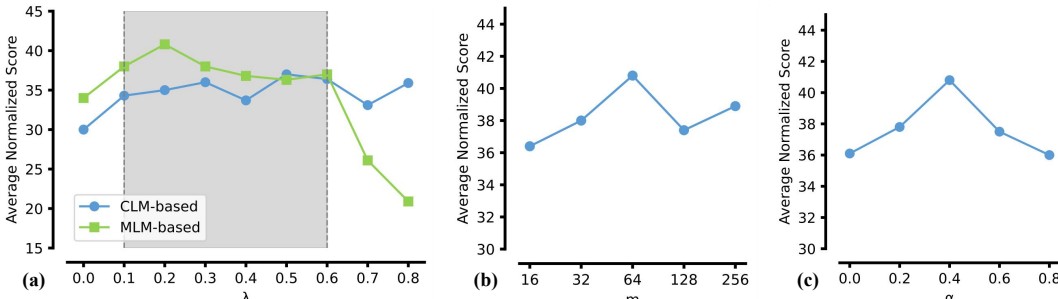

Figure 7: Effect of hyperparameters. (**a**) The weight of response modeling loss $\lambda$. (**b**) The number of repeated model ID tokens $m$. (**c**) The fraction of curriculum masking $\alpha$.

features. However, beyond a certain threshold ($m = 64$), performance declines due to the increased difficulty of predicting extended token sequences. The limited capacity of the masked language model (MLM) leads to error accumulation during long-range prediction, which introduces noise into the latent representations and degrades routing accuracy.

### E.3 Effects of $\alpha$.

Figure 7 (c) illustrates the impact of varying the fraction of curriculum masking, $\alpha$. A moderate value of $\alpha$ (e.g., 0.4) achieves optimal performance by facilitating a smooth transition from partial to full response modeling. In contrast, excessively high values of $\alpha$ ($> 0.4$) lead to performance degradation, as an extended curriculum procedure reduces the effective training time on the final objective and results in underfitting.

## F    Further analysis

### F.1    Specialization awareness

To investigate whether Lookahead effectively leverages the specialized capabilities of heterogeneous LLMs, we analyze its routing behavior on domain-specific benchmarks. Specifically, we compare the routing proportions of the MLM-based Lookahead against a multi-label classifier (MLC) baseline on mathematical reasoning (GSM8K, MATH) and code generation (HumanEval, MBPP) tasks.

As shown in Table 4, Lookahead demonstrates strong specialization awareness. On mathematical problems, it routes significantly more queries to the top-performing models identified in our main evaluation (Table 1): InternLM-2.5-20B-Chat and Qwen2.5-Coder-7B-Instruct. This trend is even more pronounced for code generation tasks. Lookahead routes nearly all code-related queries to the coding-specialized model Qwen2.5-Coder-7B-Instruct, substantially outperforming the baseline. These results confirm that response-aware modeling enables Lookahead to better identify model specialties and make precise routing decisions tailored to task requirements.

Table 4: Routing proportions (%) of the MLM-based Lookahead vs. MLC baseline on domain-specific benchmarks.

| Candidate Models | GSM8K | | MATH | | HumanEval | | MBPP | |
|---|---|---|---|---|---|---|---|---|
| | Ours | Baseline | Ours | Baseline | Ours | Baseline | Ours | Baseline |
| Yi-1.5-34B-Chat | 4.3 | 14.0 | 0.1 | 0.3 | 0.0 | 0.0 | 0.0 | 0.0 |
| InternLM-2.5-20B-Chat | 53.7 | 50.7 | 75.6 | 72.8 | 0.0 | 0.0 | 0.0 | 0.8 |
| Phi-3-Medium-4k-Instruct | 13.9 | 10.5 | 0.6 | 0.3 | 0.0 | 5.5 | 0.0 | 0.0 |
| Llama-3.1-8B-Instruct | 0.0 | 0.0 | 0.0 | 0.0 | 0.0 | 9.8 | 0.0 | 0.0 |
| Qwen2.5-Coder-7B-Instruct | 28.1 | 24.8 | 23.7 | 26.6 | 100.0 | 84.8 | 100.0 | 99.2 |

### F.2    Performance across query difficulty levels

To analyze how Lookahead performs on queries of varying difficulty, we categorize test samples based on the number of candidate LLMs that produce correct responses. We then compare the

performance of the MLM-based Lookahead against the MLC baseline across these grouped samples. For each query, we classify the outcome as a *Win* if Lookahead selects a correct model while the baseline does not, a *Loss* if the baseline succeeds and Lookahead fails, and a *Tie* otherwise.

As shown in Table 5, Lookahead demonstrates a clear advantage on more complex queries. The *Win-Loss* margin is most pronounced (+2.2%) for the most challenging category (where only one model produces a correct response). This suggests that by modeling the latent semantics of potential responses, Lookahead is better equipped to identify the single capable model among many for hard queries, where query-only methods often fail to capture the subtle cues needed for accurate routing.

Table 5: Performance of MLM-based Lookahead vs. MLC baseline on queries grouped by the number of correct candidate responses.

| # Correct Candidate Responses | Win | Tie | Loss | Win $-$ Loss |
|---|---|---|---|---|
| 1 | 9.9% | 82.5% | 7.7% | 2.2% |
| 2 | 7.2% | 86.0% | 6.8% | 0.4% |
| 3 | 7.0% | 87.6% | 5.4% | 1.6% |
| 4 | 4.2% | 91.5% | 4.4% | -0.2% |

### F.3 Performance-efficiency tradeoff

To evaluate the performance-efficiency tradeoff achieved by Lookahead, we compare its MLM-based implementation against three baselines: (i) a multi-label classifier (MLC), (ii) the best single candidate model, and (iii) an ensembling method where a reward model selects the best candidate response.

As shown in Table 6, both model ensembling and routing surpass the best single model by a large margin in performance. However, model ensembling incurs a heavy computational cost by generating with 83.8B parameters for each query, while MLC and Lookahead both reduce this cost to only about 21%, which demonstrates that Lookahead achieves an accuracy-efficiency tradeoff on par with other routing methods and surpasses single model inference or model ensembling.

Table 6: Average normalized performance score and activated parameters to generate the final response for each query.

| Method | Performance | Parameters |
|---|---|---|
| Best Single Model (Qwen2.5-Coder-7B-Instruct) | 13.4 | 7.6B |
| Model Ensembling (Reward Model Select) | 48.8 | 83.8B |
| Routing Baseline (MLC) | 34.0 | 17.51B |
| Lookahead | 40.8 | 17.37B |

### F.4 Scalability

To assess the scalability of Lookahead as the candidate model pool expands, we analyze both theoretical computational overhead and empirical performance trends.

The inference-time overhead of Lookahead stems from processing additional model identifier (MID) tokens. For the CLM-based variant, since the router only needs the hidden state at the MID token to form the response representation without performing actual text generation, all $T$ MID tokens can be concatenated into a single input sequence $(x \parallel \text{MID}_1 \parallel \ldots \parallel \text{MID}_T)$ with a modified attention mask to prevent cross-interference. This design requires only a single forward pass, incurring an overhead of $T$ extra tokens. For the MLM-based variant, the input sequence includes $m$ repeated MID tokens for each of the $T$ models, resulting in $m \times T$ additional tokens processed in one joint forward pass. In both cases, the overhead grows linearly with the number of candidates $T$. However, given that the router backbones are small and $T$ is typically modest, this added cost is negligible compared to the computational cost required for response generation by any candidate LLM.

We empirically validate scalability by expanding the candidate pool size from 3 to 8 models. The first five models are included in order as listed in Table 1, while details on three additional candidates are provided in Table 7. Table 8 reports normalized scores on representative benchmarks, along with router GFLOPs and latency (measured on an NVIDIA RTX 3090). The results show that routing

performance initially improves when expanding the candidate pool from three to five models, as the inclusion of stronger and complementary specialists enhances coverage of diverse task requirements. However, further expansion to eight models leads to a slight performance decline, primarily due to the introduction of weaker or redundant models that increase routing noise without contributing meaningful capability. This suggests that a small set of high-quality, complementary models is sufficient to form an effective candidate pool. In terms of computational overhead, CLM-based Lookahead incurs only approximately 4.6% additional cost over its baseline when $T = 5$, while MLM-based Lookahead remains below 5% relative to generating just the first token from even the smallest candidate model (Qwen2.5-Coder-7B-Instruct, requiring 1810 GFLOPs). As the number of candidates increases, both implementations exhibit only modest growth in latency. Although the MLM-based variant scales faster in FLOPs due to its $m \times T$ token expansion, its absolute overhead remains negligible compared to the cost of autoregressive decoding in LLMs. These results confirm that Lookahead's design balances improved routing decisions with minimal added complexity, even as candidate pools grow larger.

Table 7: Comparison of Additional Candidate Models

| Hugging Face Model ID | Parameters | AlpacaEval | MATH | HumanEval |
|---|---|---|---|---|
| deepseek-ai/deepseek-coder-6.7b-base [18] | 6.7B | 2.7 | 4.8 | 75.6 |
| meta-llama/Llama-3.2-3B-Instruct [16] | 3.2B | 21.9 | 39.6 | 54.9 |
| Qwen/Qwen2.5-1.5B-Instruct [34] | 1.5B | 10.4 | 49.9 | 58.5 |

Table 8: Performance, router computational cost, and latency as the number of candidate models increases.

| Method | #Models | AlpacaEval | MATH | HumanEval | Computational Cost / GFLOPs | Latency / ms |
|---|---|---|---|---|---|---|
| *CLM-based* | | | | | | |
| MLC | 3 | 39.0 | 62.2 | 75.6 | 18.62 | 28.4 |
| | 5 | 39.4 | 62.2 | 85.4 | 18.62 | 28.0 |
| | 8 | 37.9 | 62.2 | 87.2 | 18.62 | 28.0 |
| Lookahead | 3 | 37.6 | 62.2 | 72.6 | 19.04 | 28.4 |
| | 5 | 37.8 | 62.2 | 87.2 | 19.47 | 28.4 |
| | 8 | 37.9 | 62.3 | 87.2 | 20.11 | 29.6 |
| *MLM-based* | | | | | | |
| MLC | 3 | 36.2 | 62.2 | 72.6 | 18.52 | 19.5 |
| | 5 | 38.5 | 62.0 | 83.5 | 18.52 | 19.6 |
| | 8 | 38.4 | 62.1 | 70.1 | 18.52 | 19.1 |
| Lookahead | 3 | 39.1 | 62.2 | 74.4 | 61.79 | 19.9 |
| | 5 | 40.0 | 61.9 | 87.2 | 90.49 | 20.9 |
| | 8 | 39.0 | 62.4 | 87.2 | 133.54 | 22.0 |

## G  Case study

To further investigate areas for improvement, we conduct a study on cases where Lookahead fails to select the best candidate model. We observed that the MLM-based Lookahead shows less advantage on mathematical problems compared to instruction-following tasks. This appears to be because our current implementation models differences only in the first $m$ tokens of responses. As an example from GSM8K shown in Table 9, in mathematical problems, models tend to restate the problem before solving, resulting in similar beginnings across different models. This similarity in initial tokens leads to overly similar latent representations that can mislead the router. This example highlights a key opportunity for improvement. A promising optimization is to adaptively identify the most informative spans among responses, rather than uniformly focusing on the prefix.

Table 9: A case where Lookahead chooses the response from Model 3, while only Model 4 indeed gives the correct answer.

| Model ID | Response |
|---|---|
| 1 | To determine how many years it will take before Carlos starts earning money on the ... |
| 2 | To determine how many years it will take for Carlos to start earning money on his ... |
| 3 | The cost to plant the tree is $90. \nEach year it will grow 7 lemons, which he can ... |
| 4 | To find the number of years it will take before Carlos starts earning money on the ... |
| 5 | To determine how many years it will take for Carlos to start earning money on his ... |

## H   Limitations

There are three potential limitations to our work. First, our current approach focuses solely on performance optimization and does not explicitly account for cost trade-offs between large and small models. Second, while the proposed response modeling task is compatible with various routing objectives through its dual-task formulation, we have not yet investigated its integration with alternative loss functions such as Kullback-Leibler divergence [26] or contrastive losses [9]. Third, if the reward model used during training fails to detect biased or factually incorrect outputs, the router may learn to favor such responses, inadvertently amplifying harmful content. To mitigate this risk, future work could integrate ensembles of diverse reward models or incorporate fairness-aware evaluation metrics into the routing objective to improve robustness against biased or unsafe content.

