# OpenReview forum: "Lookahead Routing for Large Language Models"
_NeurIPS.cc/2025/Conference — NeurIPS 2025 poster_

### Official Review · Reviewer_1mHC · 2025-06-28

**Clarity:** 3
**Significance:** 1
**Originality:** 1
**Rating:** 3
**Confidence:** 3

**Summary:**

Lookahead is a routing framework for large language models (LLMs) designed to improve the efficiency of multi-model systems. Current routing methods primarily use the input query to decide which LLM to use, often leading to suboptimal decisions because they don't consider the potential output. Lookahead addresses this by predicting the latent representations of potential model outputs, which allows for more informed routing decisions without the need for a full inference. The framework is implemented using both causal and masked language models. Across seven public benchmarks, Lookahead consistently surpasses existing routing methods, achieving an average performance gain of 7.7% over the state-of-the-art.

**Questions:**

Nowadays, users are turning to reasoning/thinking models for better performance on complex tasks. Conducting an evaluation of Lookahead on reasoning models would be interesting.

**Ethical Concerns:**

["NO or VERY MINOR ethics concerns only"]

**Final Justification:**

The rebuttal addresses some of my concerns, but I still find the proposed approach to be incremental and model routing will likely become a less relevant problem as foundation models evolve.

**Limitations:**

Introducing an additional routing classifier complicates the online serving system. Including such a discussion in the paper would help readers understand the trade-offs involved.

**Quality:**

2

**Strengths And Weaknesses:**

### Strengths

1. The proposed Lookahead method consistently outperforms existing routing baselines across seven diverse benchmarks, including instruction following, mathematical reasoning, and code generation.
2. The method is straightforward and easy to implement, which is good for practical applications.

### Weaknesses

1. Limited Novelty: the proposed method basically adds an auxiliary task to train routing classifier by predicting LLM's latent representations. Although empirically effective, the novelty is quite limited and may not suffice for a top-tier conference.

2. Relevance of Routing as Models Evolve: As LLMs become more powerful and general-purpose, the need for routing may diminish. Larger models are almost always better than smaller ones across a wide spectrum of tasks, which raises the question of whether routing will still be necessary in the future. This paper only considers task performance for model routing, but I think a more appropriate setting is to take into account the cost of inference. For example, for easy queries, smaller models are often sufficient while being much cheaper to run.

3. Including evaluation on recent reasoning models would be beneficial.

---

> ### Author Rebuttal · Authors · 2025-07-31
>
> Thanks for the valuable comments. We really appreciate your efforts to help us improve our paper. We carefully addressed your concerns below and sincerely hope that our reply resolves your concerns. Please let us know if you have any follow-up questions.
>
> > **Q1: Concern about novelty.**
>
> A1: We would like to highlight that the core novelty of our work lies in addressing a critical **limitation of existing routing methods: their inability to leverage potential model outputs during routing decisions**. Current approaches rely solely on input features, neglecting the contextual nuances and implicit intent that emerge during response generation. Our proposed **response-aware router** introduces a paradigm shift by:
>
> 1. **Predicting latent representations** of candidate model outputs (via MLM/CLM-based exploration) to "foresee" their semantic contributions.
>
> 2. **Enabling informed routing** without full inference, which is particularly effective for complex/ambiguous queries requiring deeper semantic understanding.
>
> This framework provides a new perspective for routing research, as evidenced by the significant performance gains across diverse tasks.
>
> > **Q2: Regarding the necessity of the routing task.**
>
> A2: While LLMs continue to evolve, we emphasize that **no single model consistently excels across all scenarios**—different models often have complementary strengths (e.g., domain-specific expertise vs. general-purpose capabilities). Routing remains crucial to harness these strengths effectively, as supported by recent studies (e.g., [1][2]).
>
> [1] Harnessing the Power of Multiple Minds: Lessons Learned from LLM Routing. ArXiv 2024.
>
> [2] Routing to the Expert: Efficient Reward-guided Ensemble of Large Language Models. NAACL 2024.
>
> > **Q3: Regarding inference cost in modeling.**
>
> A3: Thank you for raising this important point. While we agree that cost-aware routing is critical for practical deployments, we need to clarify that model **routing research can generally be categorized into two main directions: (1) improving the prediction of candidate model performance (e.g., [1][2][3]), and (2) balancing multiple objectives such as performance, cost, and latency (e.g., [4][5][6]). Our work falls into the first category**, focusing on incorporating candidate model responses to enhance performance prediction rather than explicitly addressing trade-offs between performance and cost.
>
> That said, we believe our approach could complement multi-objective routing methods to improve overall effectiveness. Accurate performance prediction is foundational to any cost-performance trade-off; thus, our response-aware framework could be integrated with approaches like MixLLM [6] to extend routing objectives beyond pure performance—incorporating factors such as response length, inference latency, and computational cost. These objectives could be balanced using adjustable weighting factors tailored to specific requirements or resource constraints.
>
> We appreciate your valuable suggestion and will include a discussion on this potential extension in the revised manuscript.
>
> [1] Routing to the Expert: Efficient Reward-guided Ensemble of Large Language Models. NAACL 2024.
>
> [2] RouterDC: Query-based Router by Dual Contrastive Learning for Assembling Large Language Models. NeurIPS 2024.
>
> [3] SMOOTHIE: Label Free Language Model Routing. NeurIPS 2024.
>
> [4] Fly-Swat or Cannon? Cost-Effective Language Model Choice via Meta-Modeling. WSDM 2024.
>
> [5] RouteLLM: Learning to Route LLMs from Preference Data. ICLR 2025.
>
> [6] MixLLM: Dynamic Routing in Mixed Large Language Models. NAACL 2025.
>
> > **Q4: Regarding generalization over reasoning models.**
>
> A3: We acknowledge that evaluating reasoning models could further illustrate the generalization capability of our method. However, we emphasize that **our approach has already been thoroughly tested across seven diverse benchmarks**, including instruction following, math, and code. Our method consistently outperforms baselines across all these tasks, demonstrating robust generalization capabilities.
>
> While the inference time for reasoning models on our hardware (RTX 3090) is prohibitively long, making it difficult to complete additional experiments within the rebuttal timeline, we are actively running these experiments and will share results during the discussion period.

---

> > ### Comment · Reviewer_1mHC · 2025-08-04
> >
> > Thanks for the detailed response! I will raise the score to 3, but I'd like to re-emphasize some personal beliefs that authors may disagree:
> >
> > 1. Model routing will become less relevant as foundation models evolve. A similar circumstance is that meta search engines, which try to combine multiple search results, never gained enough popularity among users.
> > 2. Predicting latent representations is nothing more than an auxiliary multi-task objective, if we have enough training data, the performance gains will likely diminish.

---

> > > ### Author Response · Authors · 2025-08-05
> > >
> > > Thank you for your thoughtful feedback. To address your concerns more comprehensively, we provide detailed responses with evidence below:
> > >
> > > > **Q1: Relevance of model routing as models evolve.**
> > >
> > > A1: While we acknowledge that foundation models are evolving and becoming more versatile, we want to emphasize that model routing remains highly relevant, supported by both current empirical evidence and enduring constraints that will persist regardless of advancements in foundation models.
> > >
> > > 1. **Current State:** While general-purpose foundation models have made significant progress, no single model achieves universal expertise across all domains. Task-specific optimization continues to produce specialized models (e.g., Qwen3-coder, Qwen2.5-Math) that outperform generalist alternatives in their respective areas, underscoring **the continued importance of model routing**. The increase in routing research at premier conferences (e.g., [1][2][3][4][5]) further highlights the ongoing relevance of this area. Additionally, platforms like Not Diamond and LangChain leverage routing to integrate specialized models effectively.
> > >
> > > 2. **Future Outlook:** Even as foundation models grow more capable, **structural limitations ensure the continued relevance of intelligent routing mechanisms**. Industries handling proprietary or sensitive data (e.g., financial systems, medical databases) face legal, ethical, and competitive barriers preventing direct integration into public foundation models. These constraints necessitate domain-specific solutions that rely on robust model-routing architectures to integrate specialized knowledge seamlessly within broader AI ecosystems.
> > >
> > >
> > > [1] BEST-Route: Adaptive LLM Routing with Test-Time Optimal Compute. ICML 2025.
> > >
> > > [2] Capability Instruction Tuning: A New Paradigm for Dynamic LLM Routing. AAAI 2025.
> > >
> > > [3] RouteLLM: Learning to Route LLMs from Preference Data. ICLR 2025.
> > >
> > > [4] RouterDC: Query-Based Router by Dual Contrastive Learning for Assembling Large Language Models. NeurIIPS 2024.
> > >
> > > [5] AutoMix: Automatically Mixing Language Models. NeurIPS 2024.
> > >
> > > > **Q2: Regarding the novelty of latent response representation prediction.**
> > >
> > > A2: To clarify, the response representation prediction in Lookahead is **not merely an auxiliary task but a core innovation that fundamentally transforms the routing** **paradigm**.
> > >
> > > 1. **Beyond an auxiliary task:** Traditional auxiliary tasks serve as regularizers to enhance representation robustness. In contrast, Lookahead's response modeling is integral to routing decisions. Our preliminary study in Section 1 clearly demonstrates that routing performance critically depends on access to response semantics, which our latent representations directly encode.
> > >
> > > 2. **Empirical evidence:** As shown in Section 5.4 and Appendix D (Fig. 5a, 6a), Lookahead's **performance gains persist with increasing** **training data**. This indicates that response modeling captures intrinsic task-specific signals rather than acting as a simple regularizer.
> > >
> > > In essence, Lookahead's response-aware framework shifts routing from a query-prediction paradigm to a response-anticipation paradigm, where **latent representations serve as the causal bridge between input and optimal model selection**. This is fundamentally different from auxiliary multi-task learning.
> > >
> > > We hope these clarifications address your concerns. Please feel free to reach out if you have any follow-up questions.

---

### Official Review · Reviewer_uF3D · 2025-07-01

**Clarity:** 4
**Significance:** 3
**Originality:** 3
**Rating:** 5
**Confidence:** 3

**Summary:**

This paper introduces "Lookahead," a novel framework for routing queries within multi-model LLM systems. Traditionally, these systems route queries based purely on the input, which often falls short for complex or ambiguous requests where the true intent only emerges during the LLM's response generation. Lookahead tackles this limitation by training the router to "foresee" what a given LLM might output. It achieves this not by performing full inference, but by predicting the latent representations of these potential responses. These predicted latent features then directly guide the model selection, leading to more accurate and informed routing decisions without the computational cost of running all candidate LLMs. The framework is instantiated in two variants: one using a causal language model and another a masked language model. Both are trained with a joint objective that combines the routing classification loss with an auxiliary reconstruction loss for the response prediction, ensuring the latent features are meaningful. Extensive empirical evaluations across seven benchmarks consistently show Lookahead outperforming existing query-only routing baselines, delivering gains in both performance and data efficiency.

**Questions:**

1. Could you provide a more detailed breakdown of Lookahead's performance improvements, perhaps per-LLM or per-task category? For instance, does Lookahead offer disproportionate benefits when routing queries to smaller, more specialized models, or does it shine particularly on ambiguous queries where query-only methods often fail? A qualitative error analysis on routing mistakes made by Lookahead versus baselines would be insightful. What specific types of queries still pose a challenge for your framework, and what might be the underlying reasons?
2. While efficiency is a stated benefit, can you provide concrete numbers for the inference latency of the Lookahead router itself? How much does the latent prediction module add to the overall routing decision time compared to a baseline query-only classifier? Additionally, how does the computational complexity of the Lookahead prediction model scale with an increasing number or diversity of candidate LLMs?
3. Real-world LLM systems are rarely static; new models are integrated, and existing ones are frequently updated or fine-tuned. How robust is Lookahead to such changes? If a new LLM is added or an existing one receives a significant update, would the Lookahead router need to be retrained entirely?

**Ethical Concerns:**

["NO or VERY MINOR ethics concerns only"]

**Final Justification:**

I maintain my positive view towards this paper after the rebuttal. I did not give even higher score as the significance and the originality are not that exceptional.

**Limitations:**

While the authors have clearly articulated technical limitations, they have not adequately addressed potential negative societal impact.
Authors might consider the following:
1. How might routing decisions, if unchecked, amplify biases present in the training data of the router or the LLMs themselves? Discussion on potential bias amplification or mitigation strategies (e.g., fairness-aware routing metrics) is warranted.
2. Consider how the router's choices might impact the spread of misinformation or factual accuracy, especially if it prioritizes models prone to certain types of errors.

**Quality:**

3

**Strengths And Weaknesses:**

Strengths:
1. The central idea of integrating "generative foresight" via latent response prediction into LLM routing is novel. It effectively bridges the performance gap between computationally cheap query-based routing and the otherwise prohibitive cost of full-response-based routing. The specific architectural instantiations with CLM and MLM, particularly the joint prediction strategy and the curriculum masking in the MLM variant, highlight innovative design.
2. More efficient and accurate routing translates directly into reduced operational costs and enhanced user experience in multi-model LLM deployments. This work also deepens our understanding of what constitutes truly effective routing signals, underscoring that incorporating response-level information—even in a latent form—is crucial. This could very well spur a new line of research into advanced "lookahead" mechanisms across various LLM-related tasks.
3. The paper is technically robust. The claims are thoroughly supported by comprehensive empirical evaluations across a diverse set of benchmarks. The ablation studies are strong; they meticulously isolate and demonstrate the individual contributions of key components like the response modeling objective, curriculum masking, and joint prediction, thereby validating the core design choices.
4. The submission is generally very well-written and structured. The problem is clearly defined, and the proposed solution is articulated with enough detail for an expert to grasp the mechanics. The distinction between the CLM and MLM variants is clear, and the joint training objective is presented concisely. The figures are illustrative and support the narrative effectively.

Weaknesses:
1. While the paper evaluates across various benchmarks, a more granular breakdown of Lookahead's performance across individual LLMs within the 5-model system would add depth. Understanding if Lookahead particularly benefits routing to specialized models versus general-purpose ones, or for specific query types (e.g., highly ambiguous rounted to larger model? or code / logical reasoning related tasks routed to coder model), would offer richer insights.
2. The paper asserts low computational cost, but a more explicit quantification of the inference latency of the Lookahead router itself (compared to a simpler classifier baseline) would be beneficial. Pinpointing the exact added latency from the latent response prediction module would provide a more complete practical picture.

---

> ### Author Rebuttal · Authors · 2025-07-31
>
> We sincerely thank you for the detailed and positive comments. We take all comments seriously and do our best to address every concern raised. Please let us know if you have any follow-up questions.
>
> > **Q1: Regarding a more granular analysis of Lookahead's performance.**
>
> A1: Thank you for this thoughtful suggestion. We agree that a detailed analysis of Lookahead's routing patterns would provide deeper insights into its effectiveness. To address this, we conducted an evaluation on math and code benchmarks (GSM8K, MATH, HumanEval, MBPP), comparing Lookahead with a query-only baseline where response-aware modeling is removed.
>
> Our findings highlight that **Lookahead demonstrates strong specialization awareness in its routing decisions**:
>
> - For math queries (GSM8K and MATH), Lookahead frequently routes to InternLM-2.5-20B-Chat and Qwen2.5-Coder-7B-Instruct—the top two models for math tasks as identified in Table 1—significantly more often than the baseline.
>
> - For code queries (HumanEval and MBPP), it consistently selects Qwen2.5-Coder-7B-Instruct—the coding-specialized model—with higher accuracy compared to the baseline.
>
>
> These results confirm that response-aware modeling enables Lookahead to better identify model specialties and make precise routing decisions tailored to task requirements. We will incorporate these analyses into the revised manuscript to further demonstrate our method’s strengths.
>
> |Candidate Models|GSM8K (Lookahead vs Baseline)|MATH (Lookahead vs Baseline)|HumanEval (Lookahead vs Baseline)|MBPP (Lookahead vs Baseline)|
> |---|---|---|---|---|
> |Yi-1.5-34B-Chat|4.32% vs 13.95%|0.12% vs 0.3%|0.00% vs 0.00%|0.00% vs 0.00%|
> |InternLM-2.5-20B-Chat|53.68% vs 50.72%|75.56% vs 72.8%|0.00% vs 0.00%|0.00% vs 0.78%|
> |Phi-3-Medium-4k-Instruct|13.87% vs 10.54%|0.58% vs 0.3%|0.00% vs 5.49%|0.00% vs 0.00%|
> |Llama-3.1-8B-Instruct|0.00% vs 0.00%|0.00% vs 0.00%|0.00% vs 9.76%|0.00% vs 0.00%|
> |Qwen2.5-Coder-7B-Instruct|28.13% vs 24.79%|23.74% vs 26.6%|100.00% vs 84.76%|100.00% vs 99.22%|
>
> > **Q2: Regarding qualitative error analysis.**
>
> A2: To analyze Lookahead's errors qualitatively, we categorized test samples based on the number of candidate LLMs capable of providing correct responses (ranging from 0 to 5). We then evaluated Lookahead’s win rate against baselines on samples with 1–4 correct responses, where:
>
> - **Win:** Lookahead routed to a correct response while baselines failed.
>
> - **Loss:** Baselines succeeded while Lookahead did not.
>
> - **Tie:** Both succeeded or failed simultaneously.
>
>
> Our findings show that **Lookahead performs better on more complex queries**, particularly those where only a few models can generate correct responses. This highlights the strength of response-aware modeling in understanding and routing for challenging tasks.
>
> |# Correct candidate responses|Win|Tie|Loss|Win - Loss|
> |---|---|---|---|---|
> |1|9.9%|82.5%|7.7%|2.2|
> |2|7.2%|86.0%|6.8%|0.4|
> |3|7.0%|87.6%|5.4%|1.6|
> |4|4.2%|91.5%|4.4%|-0.2|
>
> To further investigate areas for improvement, we observed that MLM-based Lookahead is less effective on mathematical problems compared to instruction-following tasks. This limitation arises because our current implementation models differences only in the first $m$ tokens of candidate responses. For example, in GSM8K math problems, most models restate the problem before solving it—resulting in highly similar initial tokens across different models and overly similar latent representations that can mislead routing decisions.
>
> A promising optimization would be to adaptively model token positions with the greatest semantic differences among candidate responses rather than focusing fixedly on initial tokens. We will include these results and analyses in the revised manuscript along with potential future directions for improvement.
>
> | | |
> |---|---|
> |Query|`Carlos is planting a lemon tree. The tree will cost $90 to plant. Each year it will grow 7 lemons, which he can sell for $1.5 each. It costs $3 a year to water and feed the tree. How many years will it take before he starts earning money on the lemon tree?`|
> |Response 1|`To determine how many years it will take before Carlos starts earning money on the lemon tree, we need to calculate ...`|
> |Response 2|`To determine how many years it will take for Carlos to start earning money on his lemon tree, we need to calculate ...`|
> |Response 3|`The cost to plant the tree is $90.\nEach year it will grow 7 lemons, which he can sell for $1.5 each.\nSo, he will earn 7 * $1.5 = $10.5 from selling the lemons each year ...`|
> |Response 4|`To find the number of years it will take before Carlos starts earning money on the lemon tree, we need to consider ...`|
> |Response 5|`To determine how many years it will take for Carlos to start earning money on his lemon tree, we need to calculate ...`|
> |Predicted Target|Model 3|
> |Ground-Truth Target|Model 4|
>
> > **Q3: Regarding the computational complexity with more candidate models.**
>
> A3: Thank you for raising this important point. We acknowledge that response-aware modeling in Lookahead introduces additional complexity compared to a query-only baseline. However, this overhead is minimal relative to LLM inference and is justified by the significant performance gains it delivers. To clarify, **response-aware modeling predicts latent representations of candidate responses and uses these predictions to guide model selection without requiring full inference**. Below, we detail the computational efficiency of our approach:
>
> - CLM-based implementation:
>
>     - During training, the model generates responses using input sequences ($x\|MID_t$), where $MID_t$ represents a model identifier. At inference time, only the hidden state at $MID_t$ is used to predict response quality—avoiding actual generation.
>
>     - To further optimize efficiency, all model IDs can be packed into a single sequence ($x\|MID_1\|\dots\|MID_T$) with modified attention masks ensuring proper isolation between identifiers. This requires just one forward pass with an additional $T$ tokens compared to a simple classifier.
>
> - MLM-based implementation:
>
>     - All candidate responses are reconstructed jointly in one forward pass instead of multiple inferences.
>
>     - While introducing $m*T$ extra tokens (where $m$ is fixed per response), experiments show that small values of $m$ (16–64 tokens) yield substantial performance improvements (as shown in Figure 7(b)).
>
>
> Experimental results demonstrate:
>
> -  With five candidate models:
>
>     -  The CLM-based Lookahead adds only ~4.6% computational cost compared to a query-only router while maintaining high routing accuracy.
>
>     -  The MLM-based variant incurs slightly higher costs but remains under 5% of the computation required by Qwen2.5-Coder-7B-Instruct—the smallest LLM in our pool—to generate just its first token for queries averaging 88 tokens.
>
> -  As the number of candidates increases from three to eight:
>
>     -  Both variants exhibit slow growth in latency due to efficient batching mechanisms.
>
>     -  While MLM-based Lookahead scales faster because it introduces $m*T$ extra tokens per model, its overall cost remains negligible compared to LLM decoding overheads.
>
>
> These findings confirm that Lookahead's design balances improved routing decisions with minimal added complexity even as candidate pools grow larger. We will incorporate these analyses and experimental results into the revised paper for clarity and completeness.
>
> |**Model**|**#Models**|**Computational Cost /** **GFLOPs**|**Inference Latency / ms**|
> |---|---|---|---|
> |MLC (CLM-based)|3|18.62|28.4|
> | |5|18.62|28.0|
> | |8|18.62|28.0|
> |Lookahead (CLM-based)|3|19.04|28.4|
> | |5|19.47|28.4|
> | |8|20.11|29.6|
> |MLC (MLM-based)|3|18.52|19.5|
> | |5|18.52|19.6|
> | |8|18.52|19.1|
> |Lookahead (MLM-based)|3|61.79|19.9|
> | |5|90.49|20.9|
> | |8|133.54|22.0|
> |Qwen2.5-Coder-7B-Instruct (Generating the First Token)|1|1810|-|
>
> > **Q4: Regarding retrain Lookahead when the candidate model pool is updated.**
>
> A4: We acknowledge that significant updates to candidate models or the addition of new LLMs would require retraining Lookahead. However, we want to clarify that **this is not a limitation unique to our approach but rather a fundamental characteristic of most routing frameworks**. Routing methods inherently rely on learning performance characteristics from training queries, and any substantial changes in model capabilities or the introduction of new models necessitate updating the router to adapt to these shifts.
>
> That said, routers are typically lightweight and designed for rapid retraining. For example, our MLM-based Lookahead implementation (using ModernBERT-base) can be fully retrained in under one hour on a single NVIDIA RTX 3090 GPU. This low computational cost ensures that Lookahead remains practical even in dynamic environments where periodic model updates occur.
>
> > **Q5: Regarding the risk of amplifying biases and misinformation.**
>
> A5: Thank you for highlighting this limitation. We acknowledge the inherent risk that routing decisions could amplify biases or misinformation if reward models fail to detect biased or erroneous outputs from LLMs, potentially propagating issues present in the training data.
>
> To mitigate this, we propose deploying **ensemble methods** **with diverse reward models**, which can enhance detection accuracy and reduce the likelihood of misclassifying biased or incorrect responses as high-quality. Furthermore, incorporating fairness-aware metrics and robust evaluation criteria into our framework could provide additional safeguards against these risks.
>
> We will include a detailed discussion on these potential risks and mitigation strategies in the revised manuscript.

---

> > ### Comment · Reviewer_uF3D · 2025-08-04
> >
> > Thanks authors for the detailed reply, I have read and decided to keep my positive score.

---

### Official Review · Reviewer_AuFa · 2025-07-02

**Clarity:** 2
**Significance:** 3
**Originality:** 3
**Rating:** 5
**Confidence:** 3

**Summary:**

This paper introduces a routing framework called Lookahead for multi-LLM systems. Different from traditional routing methods, which rely solely on the input query to select the most appropriate model, the key design of Lookahead addresses this by predicting the latent representations of potential model outputs and using them to guide routing decisions. The authors implement two variants of Lookahead using causal and masked language models and evaluate them across seven benchmarks covering instruction-following, mathematical reasoning, and code generation. The results show consistent improvements over existing routing methods, with an average performance gain of 7.7% over the existing state-of-the-art.

**Questions:**

* The paper could benefit from a deeper qualitative analysis of where Lookahead fails or underperforms.
* Why do you use binary cross-entropy (L177 & Eq. 4) for a multi-class classification (multi-model routing) problem? As mentioned in L131-132 & Eq 1, only one model will be finally selected for inference.
* How to set the value of $m$ in Eq. 7? Should $m$ be large enough to cover all possible response lengths? If yes, what is the overhead of the routing model like?
* In L229-232,  the UltraFeedback benchmark relies on a reward model (Skywork-Reward-Gemma-2-27B-v0.2) for evaluation. Could it introduce bias or noise into the performance metrics? Especially, I notice that changing this reward model to another (Skywork-Reward-Gemma-27B) can result in significantly different results according to the "Reward Model Select" row in  Table 1.
* In L211, it is inappropriate to mention "theoretical justification" unless your analysis is genuinely supported by theories.

**Ethical Concerns:**

["NO or VERY MINOR ethics concerns only"]

**Final Justification:**

The rebuttal has addressed my earlier concerns on the experimental setting, method advantage, and training objectives.

**Limitations:**

yes

**Paper Formatting Concerns:**

N/A.

**Quality:**

3

**Strengths And Weaknesses:**

## Strengths

* The idea behind Lookahead is novel and reasonable. Predicting the latent representations of target LLMs may provide a meaningful supervision signal for the router to select models.
* This paper offers an in-depth examination of technical implementations. For instance, both causal language models (CLM) and masked language models (MLM) are considered as the representation predictor, and an ablation study on key designs (Sec. 5.3) is provided.
* This paper is well-organised and easy to follow. Clear background and problem formulation are presented for broader appeal.

## Weaknesses

* Insufficient justification for the experimental setting. In L252-L253, it is mentioned that "For a fair comparison, the classifier-based baselines are reimplemented using the same backbones as Lookahead." Why not align the backbone choice with existing methods? Reimplementing existing methods has a risk of unfair comparison. If these methods all use different backbones, it would be comprehensible to align the best one.
* Unclear strengths over baseline methods. In Table 1, it is noticeable that general pretrained embedding models can perform as well as or even better than the proposed Lookahead. Does Lookahead have any other strengths over them?
* Only accuracy metrics are reported, while the efficiency metrics are missing. As mentioned in the Abstract, a crucial motivation is to "improve the efficiency of multi-model systems". It is advisable to incorporate a quantitative analysis of the accuracy-efficiency tradeoff.

---

> ### Author Rebuttal · Authors · 2025-07-31
>
> Thank you for your detailed review and insightful questions on our paper. Below, we address your concerns in detail. Please let us know if you have any follow-up questions.
>
> > **Q1: Regarding the reason for reimplementing baselines.**
>
> A1: To clarify, the primary reason for reimplementing baseline methods is the absence of publicly available training datasets that include model responses, which are essential for our response-aware routing framework. Since we needed to implement all methods on our customized training set, we chose to use stronger backbones and keep them consistent across all approaches to ensure a fair comparison.
>
> > **Q2: Regarding the strengths of Lookahead over pretrained embedding based methods.**
>
> A2: To clarify, while certain pretrained embedding-based methods demonstrate competitive results on specific benchmarks, such as kNN on MATH and SMOOTHIE on MBPP. We would like to emphasize that **response-aware modeling enables Lookahead to achieve SOTA performance not only in overall metrics but also in 5 out of these 7 individual benchmarks.** Below are three key strengths of Lookahead compared to pretrained embedding-based approaches:
>
> 1. **Predictive Latent Representations:** By forecasting latent representations of potential model outputs, Lookahead captures semantic nuances, which allows for more contextually informed routing decisions compared to embedding methods that rely solely on query similarity.
>
> 2. **Inference Efficiency:** Unlike similarity-based methods (e.g., kNN or SMOOTHIE), which compute distances between queries and training samples or cluster centers during inference—a computationally expensive step—Lookahead generates its response-aware features in a single forward pass, significantly reducing latency while maintaining high accuracy.
>
>
> > **Q3: Regarding the accuracy-efficiency tradeoff.**
>
> A3: To clarify, the statement in our abstract about "improving the efficiency of multi-model systems" refers to the **fundamental advantage of routing approaches compared to alternative multi-model strategies such as ensemble methods or cascaded inference**. As established in prior routing research [1][2], routing queries to the suitable model inherently reduces computational overhead of inference on all candidate models introduced by ensemble methods while maintaining performance advantages over using any single model alone. Our work builds upon this established efficiency foundation and focuses on improving routing accuracy.
>
> To further demonstrate that Lookahead achieves an accuracy-efficiency tradeoff on par with other routing methods and surpasses single model inference or model ensembling, we compared the MLM-based Lookahead with its multi-label classifier baseline, the best candidate model, and an ensembling method where a reward model selects the best candidate responses. As shown below, both model ensembling and routing surpass the best single model by a large margin in performance. However, **model ensembling incurs a heavy computational cost by generating with totally 83.8B parameters for each query, while MLC and Lookahead both reduce this cost to only about 21%**.
>
> |Method|Performance|Response Generation Param|
> |---|---|---|
> |Best Single Model (Qwen2.5-Coder-7B-Instruct)|13.4|7.6 B|
> |Model Ensembling (Reward Model Select)|48.8|83.8 B|
> |Routing Baseline (MLC)|34.0|17.51 B|
> |Lookahead|40.8|17.37 B|
>
> We will include these results to the updated paper.
>
> [1] Harnessing the Power of Multiple Minds: Lessons Learned from LLM Routing. ArXiv 2024.
>
> [2] Routing to the Expert: Efficient Reward-guided Ensemble of Large Language Models. NAACL 2024.
>
> > **Q4: Regarding cases where Lookahead might fail or underperform.**
>
> A4: We acknowledge that the analysis of routing errors can indeed reveal valuable insights that guide potential improvements. For example, we observed that the MLM-based Lookahead shows less advantage on mathematical problems compared to instruction-following tasks. This appears to be because our current implementation models differences only in the first $m$ tokens of responses. As an example from GSM8K shown below, **in mathematical problems, models tend to restate the problem before solving, resulting in similar beginnings across different models. This similarity in initial tokens leads to overly similar latent representations that can mislead the router**. This example highlights a key opportunity for improvement. A promising optimization is to adaptively identify the most informative spans among responses, rather than uniformly focusing on the prefix.
>
> |   |   |
> |---|---|
> |Response 1|`To determine how many years it will take ...`|
> |Response 2|`To determine how many years it will take ...`|
> |Response 3|`The cost to plant the tree is $90 ...`|
> |Response 4|`To find the number of years it will take ...`|
> |Response 5|`To determine how many years it will take ...`|
> |Predicted Target|Model 3|
> |Ground-Truth Target|Model 4|
>
> We will include these results and analysis in the revised paper.
>
> > **Q5: Regarding the reason for using BCE loss.**
>
> A5: Thank you for raising this important question. While the final routing decision selects a single model, we formulate the training process as a multi-label classification problem rather than multi-class classification following previous works [1][2][3]. Reasons for this distinction are outlined below:
>
> 1. In LLM routing, multiple models may produce high-quality responses for the same query. Using BCE loss allows us to independently evaluate each model's capability on a given query, rather than forcing an artificial exclusivity among candidate models.
>
> 2. This multi-label approach provides richer training signals compared to standard multi-class cross-entropy, as it preserves information about which models perform well rather than only identifying the single "best" model for each training example.
>
> 3. During inference, we select only one LLM with the highest predicted likelihood to generate the final response for the best performance.
>
>
> We will incorporate these clarifications into the revised paper.
>
> [1] Large Language Model Routing with Benchmark Datasets. ArXiv 2023.
>
> [2] Harnessing the Power of Multiple Minds: Lessons Learned from LLM Routing. ArXiv 2024.
>
> [3] EmbedLLM: Learning Compact Representations of Large Language Models. ICLR 2025.
>
> > **Q6: Regarding the setting and cost of $m$.**
>
> A6: Thank you for your insightful question regarding the selection of parameter m in our MLM-based implementation. We appreciate the opportunity to clarify this important design choice.
>
> Our experiments in Appendix E.2 demonstrate that $m$ does not need to cover all possible response lengths. Specifically, we found that setting $m=64$ achieves the best performance in our experimental setup, despite the fact that some responses may be longer and would consequently be truncated. This finding is supported by two key observations:
>
> - Even for tasks requiring longer responses, the initial segments often contain sufficient semantic information to distinguish between the capabilities of different candidate models.
>
> - As shown in Figure 7(b), increasing $m$ beyond 64 actually leads to performance degradation due to error accumulation during long-range prediction by the lightweight router.
>
>
> To further analyze the cost introduced by repeated model ID tokens, we compare the computational cost of Lookahead to its baseline and the smallest candidate LLM. As shown below, we acknowledge that the MLM-based **Lookahead introduces some extra computational cost. However, when compared to the computation that LLMs cost to generate the response and considering the marginal performance improvement of Lookahead, the extra computational cost is acceptable**.
>
> ||MLM-based Classifier|MLM-based Lookahead|Qwen2.5-Coder-7B-Instruct (First Token)|
> |---|---|---|---|
> |GFLOPs|18.52|90.49|1810|
>
> We will incorporate these analyses and experimental results in the revised paper.
>
> > **Q7: Regarding the risk of bias introduced to the performance metrics by the reward model.**
>
> A7: Thank you for your thoughtful comments. We address them from the following perspectives:
>
> - To clarify, UltraFeedback serves as a part of the training set instead of a test set in our experiments. Therefore, any potential bias from the reward model does not affect our performance metrics on the evaluation benchmarks.
>
> - For open-ended instruction-following tasks, we utilize widely used benchmarks (AlpacaEval-2, Arena-Hard, and MT-Bench), which utilize powerful LLMs like GPT-4-Preview-1106 as judges. These evaluation approaches have been demonstrated to align well with human preferences [1]. Additional details about our evaluation metrics are provided in Appendix B.
>
> - We appreciate you bringing our attention to the potential confusion in Table 1. We acknowledge that there was a typographical error in L240. The "Reward Model Select" method reported in the table actually uses the same reward model (Skywork-Reward-Gemma-2-27B-v0.2) that was employed for annotating our training data. This result serves as an upper bound for routing methods in open-ended tasks, providing a reference point for the maximum achievable performance under our experimental setup.
>
>
> [1] Judging LLM-as-a-judge with MT-Bench and Chatbot Arena. NeurIPS 2023.
>
> > **Q8: Regarding the paragraph heading in L211.**
>
> A8: Thank you for your valuable comment regarding the use of "theoretical justification" in line 211. We agree that the term "theoretical justification" implies a level of formal theoretical analysis that the paragraph does not provide. Since it primarily presents our design rationale and conceptual motivation, we will revise the paragraph heading to "Design Rationale" to more accurately reflect the nature of our discussion. The content of the section will be adjusted accordingly to maintain consistency with this more precise characterization.

---

> > ### Comment · Reviewer_AuFa · 2025-08-05
> > **My concerns are addressed**
> >
> > Thank you for the detailed and thoughtful rebuttal. I appreciate the clarifications provided regarding the experimental setting, method advantage, and training objectives. These responses have addressed my earlier concerns.

---

### Official Review · Reviewer_ttMU · 2025-07-03

**Clarity:** 4
**Significance:** 3
**Originality:** 3
**Rating:** 4
**Confidence:** 4

**Summary:**

This paper proposes an innovative LLM routing framework Lookahead, which effectively makes up for the lack of information in traditional pure classification routing when dealing with ambiguous or complex tasks by "foreseeing" the potential output of each candidate model when making routing decisions instead of relying solely on the input query. Specifically, the authors designed a sequence-level predictor based on the causal language model and a tag-level predictor based on the masked language model, both of which only calculate the potential representation during reasoning without the need for real decoding, significantly reducing computational overhead. The router adopts a dual objective in training: on the one hand, it performs model selection classification, and on the other hand, it reconstructs the real answer from the latent vector, so as to ensure that the latent features are rich in semantic information. The experiments cover seven benchmarks including instruction following, mathematical reasoning, and code generation. Lookahead improves by about 7.7% on average over the optimal baseline, and the MLM variant improves by double digits on the open instruction task, fully demonstrating the value of the answer perception signal. Ablation analysis shows that the routing performance drops significantly after removing the latent prediction component, while the auxiliary reconstruction objective improves the training data efficiency by more than 6 times. Overall, Lookahead provides a new idea for multi-model collaboration by cleverly combining generative foresight with efficient routing, and achieves considerable improvements in both performance and efficiency.

**Questions:**

1.How did you construct the training set for the router, especially for open-ended tasks where no single “correct” answer exists? For deterministic tasks like GSM8K or HumanEval, one can label a model’s output as correct/incorrect against ground truth. But for benchmarks like AlpacaEval or MT-Bench, did you use the pairwise comparisons or reward model scores to decide which model “performed best” on each query? For example, MT-Bench often provides a score for each model’s response; did you select the highest-scoring model per query as the target? And how did you handle ties or cases where multiple models produce equally valid outputs?

2.One of the attractive aspects of routing is reducing overall computation, so it would help to quantify Lookahead’s inference overhead relative to simpler routers. Can you provide any measurements or estimates of how much additional latency or FLOPs the Lookahead router introduces? For instance, using the MLM variant with 5 models and 64 mask tokens each, roughly what fraction of a single large model inference does the router cost? If the router is deployed with a larger pool of models (say 10+), would you anticipate any issues (e.g., BERT’s input length or CLM needing many forward passes)? In short, how does Lookahead scale in practice?

3.Did you consider any optimizations, such as early stopping for obviously inferior models or using a smaller set of candidate models after an initial screening?

4.The current router always picks the highest-quality model, which might be a very large model even for slightly easier queries. Could one extend the training to, say, include a penalty for choosing higher-cost models or a predicted computation time, so that the router could occasionally select a smaller model if the quality difference is minor?

5.The results show strong average performance, but could you share more about when Lookahead might fail or underperform? For example, did you observe any patterns in queries where the router picked a suboptimal model? Are there cases where the predicted latent representations might be misleading – e.g., perhaps for queries that none of the models can handle well, so all predicted responses look bad, and the router still has to choose one? It would be useful to know if the router has any confidence measure or if it could detect uncertainty. For instance, if all models’ predicted latents are equally poor, perhaps the system could default to a safer strategy like trying multiple models.

**Ethical Concerns:**

["NO or VERY MINOR ethics concerns only"]

**Final Justification:**

I remain 4 as my decision.

**Limitations:**

yes

**Paper Formatting Concerns:**

Correct format.

**Quality:**

3

**Strengths And Weaknesses:**

Strengths:

1.This is the first work to incorporate the content of the predicted response via latent features directly into LLM routing decisions. The idea is both original and intuitive: by “previewing” the possible outputs of each model without actually running them, the router can more accurately assess which model performs best, especially on complex queries whose true difficulty is only revealed during the generation process.

2.The paper designs a dual-task training scheme (jointly optimizing model selection and response reconstruction), which cleverly ensures the semantic alignment between the predicted latent vector and the true output.

3.The authors propose two different implementations based on an autoregressive language model and based on a masked language model, further consolidating the contribution of their work: this concept is not limited to a single architecture, but can be implemented in multiple ways.

4.The authors test on seven benchmarks with different domains and evaluation styles, demonstrating that the improvements of Lookahead are not limited to a single scenario. The performance improvements favor Lookahead on all tasks, and in most cases the advantages are significant.

Weaknesses:

1.The lookahead mechanism introduces additional complexity and overhead. Instead of being a simple classifier, the router contains either a generative component or a heavyweight encoder that needs to be executed for each candidate model. The authors do use smaller backbone models and argue that this is still efficient compared to running a full LLM. However, this still has a non-trivial inference cost.

2.The scalability of larger model pools has not been adequately tested. Five candidate LLMs were used in the experiment; it is unclear how performance and efficiency would change with dozens of models. The CLM method requires a separate forward calculation for each model, and the cost grows linearly; while the MLM method may face input length or video memory limitations if too many mask outputs are encoded at once.

3.Training the router requires response data for all candidate models, i.e., the output of each model must be generated for a set of training queries. The paper implicitly did this to create training pairs. While not uncommon in routing research, this data collection is an offline cost and may need to be repeated if the set of candidate models changes or if models are updated.

4.The authors did not explore integrating their latent-response approach with other advanced routing objectives like the Kullback–Leibler divergence-based reward distribution (ZOOTER) or contrastive learning (RouterDC).

---

> ### Author Rebuttal · Authors · 2025-07-31
>
> Thank you for your thoughtful review and valuable feedback. Below, we address your concerns in detail.
>
> > **Q1: Regarding additional complexity and overhead introduced by Lookahead.**
>
> A1: While we acknowledge that Lookahead introduces some additional complexity compared to simple classifiers, we would like to clarify that this overhead is minimal and justified by the significant performance gains it achieves. Our design specifically avoids generating proxy responses or requiring multiple forward passes for each query:
> - **CLM-Based Implementation:** At inference time, the hidden state at $MID_t$ encodes sufficient information to predict response quality—avoiding actual generation (as described in L189–L190). Furthermore, all model IDs can be packed into a single sequence ($x\|MID_1\|\dots\|MID_T$) with modified attention masks. This requires just one forward pass with an additional $T$ tokens compared to a simple classifier.
> - **MLM-Based Implementation:** All candidate responses are reconstructed jointly in one forward pass instead of performing multiple inferences, which introduces $m \times T$ extra tokens.
>
> To quantify the overhead:
> - The CLM-based Lookahead adds only ~4.6% more computational cost than a simple classifier.
> - The MLM-based variant incurs slightly higher costs but remains under 5% of the computational cost required by our smallest candidate model (Qwen2.5-Coder-7B-Instruct) to generate just its first token for queries averaging 88 tokens.
>
> These results demonstrate that both implementations maintain high efficiency while delivering superior routing accuracy. We will include these analyses and clarifications in the revised manuscript to address your concerns comprehensively.
> ||**CLM-based** **Classifier**|**CLM-based Lookahead**|**MLM-based** **Classifier**|**MLM-based Lookahead**|**Qwen2.5-Coder-7B-Instruct (Generating the First Token)**|
> |---|---|---|---|---|---|
> |Computational Cost / GFLOPs|18.62|19.47|18.52|90.49|1810|
>
> > **Q2: Regarding the scalability of Lookahead.**
>
> A2: We would like to clarify the scalability of Lookahead from three key perspectives:
> 1. **Computational Efficiency:** While we acknowledge that computational cost grows linearly with the number of candidate models, as explained in A1, Lookahead only introduces overhead for $T$ or $m\times T$ additional tokens.
> 2. **Resource Limitation:** Our MLM-based Lookahead leverages a ModernBERT-base backbone capable of handling input sequences up to 8K tokens (typically capable of 100+ candidate models), requiring just 8.4GB of video memory.
> 3. **Empirical Validation:** Due to time constraints during rebuttal preparation, conducting experiments on dozens of models was infeasible given data collection requirements. However, we expanded our existing pool to eight candidate models—and compared performance and efficiency between Lookahead and multi-label classifiers (MLC).
>
> Results show:
>    - Performance improves when expanding from three to five candidates due to stronger model additions.
>    - Performance gradually declines when expanding further (to eight candidates), as weaker models increase routing error risks.
>    - Notably, **Lookahead consistently outperforms classifier baselines**, while maintaining computational costs and latency aligned with theoretical expectations.
>
> | **Method**            | **#Models** | **AlpacaEval** | **MATH** | **HumanEval** | **Router Computational Cost /** **GFLOPs** | **Router** **Latency** **/ ms** |
> | --------------------- | ----------- | -------------- | -------- | ------------- | ------------------------------------------ | -------------------------------------- |
> | MLC (CLM-based)       | 3           | 39.0           | 62.2     | 75.6          | 18.62                                      | 28.4                                   |
> | | 5                     | 39.4        | 62.2           | 85.4     | 18.62         | 28.0                                       |
> | | 8                     | 37.9        | 62.2           | 87.2     | 18.62         | 28.0                                       |
> | Lookahead (CLM-based) | 3           | 37.6           | 62.2     | 72.6          | 19.04                                      | 28.4                                   |
> | | 5                     | 37.8        | 62.2           | 87.2     | 19.47         | 28.4                                       |
> | | 8                     | 37.9        | 62.3           | 87.2     | 20.11         | 29.6                                       |
> | MLC (MLM-based)       | 3           | 36.2           | 62.2     | 72.6          | 18.52                                      | 19.5                                   |
> | | 5                     | 38.5        | 62.0           | 83.5     | 18.52         | 19.6                                       |
> | | 8                     | 38.4        | 62.1           | 70.1     | 18.52         | 19.1                                       |
> | Lookahead (MLM-based) | 3           | 39.1           | 62.2     | 74.4          | 61.79                                      | 19.9                                   |
> | | 5                     | 40.0        | 61.9           | 87.2     | 90.49         | 20.9                                       |
> | | 8                     | 39.0        | 62.4           | 87.2     | 133.54        | 22.0                                       |
>
> We will include these analyses in the revised manuscript along with detailed experimental results supporting its scalability claims comprehensively!
>
> > **Q3: Regarding the cost of collecting training data.**
>
> A3: We would like to clarify that this challenge is not unique to our approach but rather inherent to most routing methodologies. While previous routing methods did not incorporate response content directly into router training, they still required generating these same responses to obtain scores used as training labels.
> Furthermore, Lookahead significantly mitigates this burden through remarkable data efficiency (refer to Section 5.4 for details).
>
> > **Q4: Regarding the application of other routing objectives.**
>
> A4: Thank you for this insightful comment, which highlights an important direction for validating the flexibility of our framework. Due to the limited time, we focused on integrating one representative objective (the KL divergence objective) to demonstrate the extensibility of our method.
> As shown below, **the CLM-based Lookahead achieves a 6.9% higher normalized score compared to standard ZOOTER**, while **the MLM-based variant demonstrates an even more substantial improvement of 13.9%.**
>
> |**Method**|**AlpacaEval-2**|**ArenaHard**|**MT-Bench**|**GSM8K**|**MATH**|**HumanEval**|**MBPP**|**Average**|
> |---|---|---|---|---|---|---|---|---|
> |ZOOTER (CLM)|27.6|22.2|8.4|18.8|36.2|64.7|24.8|29.0|
> |Lookahead (CLM)|27.3|19.3|10.2|22.4|37.8|67.7|32.3|31.0|
> |ZOOTER (MLM)|26.8|19.7|7.3|22.4|36.6|61.8|47.4|31.7|
> |Lookahead (MLM)|34.3|24.7|22.8|13.1|37.4|67.7|53.0|36.1|
>
> We will include these experimental results and discussions in the revised manuscript.
>
> > **Q5: Regarding the training data construction and evaluation metrics.**
>
> A5: We would like to clarify the training data construction process and evaluation metrics as follows:
>
> - **Training Set Construction:** During the creation of open-ended training data, we utilize the reward model to evaluate candidate responses. These scores are then normalized to the [0,1] interval for consistency.
> - **Metrics for Instruction-Following Benchmarks:** In benchmarks such as MT-Bench and AlpacaEval, evaluations are based on metrics like average score or win rate of model outputs across prompts.
>
> Due to space limitations, more details can be found in Section 5.1 and Appendix C.1.
>
> > **Q6: Regarding further efficiency optimization, such as early stopping and initial screening.**
>
> A6: Thank you for your valuable suggestions. We agree that exploring such optimizations could further improve efficiency. While adopting a widely used experimental setting without optimization in this paper, we will actively investigate additional optimization strategies that could work in conjunction with Lookahead, including hierarchical routing approaches and adaptive model selection techniques that could reduce inference overhead.
>
> > **Q7: Regarding the trade-off between performance and computational cost.**
>
> A7: To clarify, model routing research can generally be categorized into two main directions:
> 1. **Improving the prediction of candidate model performance** (e.g., [1][2][3]).
> 2. **Balancing multiple objectives**, such as performance, cost, and latency (e.g., [4][5][6]).
>
> Our work falls into the first category, focusing on incorporating candidate model responses to enhance performance prediction rather than explicitly addressing trade-offs between performance and cost.
>
> References:
> [1] Routing to the Expert. NAACL 2024.
> [2] RouterDC. NeurIPS 2024.
> [3] Smoothie. NeurIPS 2024.
> [4] Fly-Swat or Cannon? WSDM 2024.
> [5] RouteLLM. ICLR 2025.
> [6] MixLLM. NAACL 2025.
>
> > Q8: Regarding cases where Lookahead might fail or underperform
>
> A8: We acknowledge that analyzing failure cases can provide valuable insights to guide potential improvements. Specifically, we found that the MLM-based Lookahead shows less advantage on mathematical problems compared to instruction-following tasks. This is likely because our current implementation models differences only in the first $m$ tokens of responses. For example, as shown below, models often restate the problem before solving it, resulting in highly similar prefixes across different models, leading to overly similar latent representations that can mislead routing decisions. A promising optimization would be adaptively identifying the most informative spans among responses, rather than uniformly focusing on prefixes.
>
> 1. To determine how many years it will take ...
> 2. To find the number of years it will take ...
>
> We will include these analyses in the revised manuscript.

---

### Author Response · Authors · 2025-08-09

Dear Reviewers,

We sincerely appreciate the time and effort you have devoted to providing thoughtful reviews and valuable feedback. **We have carefully addressed your concerns in the following ways**:

- Clarified misunderstandings about Lookahead's implementation details.
- Conducted new experiments to analyze the computational efficiency, scalability and flexibility.
- Expanded our analyses with case studies and detailed explanations.

We hope these revisions and discussions have adequately addressed your concerns. As **the Author-Reviewer discussion phase ends today**, we would be grateful for any additional comments or questions that could further enhance our work. If you feel that we have satisfactorily addressed your concerns, we would **greatly appreciate it if you could kindly consider revisting your score.**

Best regards,

Authors

---

### Note · Authors · 2025-08-12

We thank the reviewers and the AC for their valuable time and insightful feedback. We are glad that reviewers recognized the novelty of integrating predicted latent response representations into LLM routing (`ttMU`, `AuFa`, `uF3D`), appreciated the comprehensive implementation with both CLM and MLM variants (`ttMU`, `AuFa`), and acknowledged the strong empirical performance across diverse benchmarks (`ttMU`, `uF3D`, `1mHC`).

During the rebuttal phase, we carefully addressed all raised concerns and made several enhancements to further strengthen the paper, which were positively recognized by most reviewers:

- **Efficiency and Scalability** (`ttMU`, `uF3D`): We conducted theoretical analyses and experiments with an increasing number of candidate models, confirming that Lookahead’s overhead growth is manageable while delivering consistent performance gains over the baseline.
- **Detailed Analyses** (`ttMU`, `AuFa`, `uF3D`): We compared Lookahead and the baseline on samples of varying difficulty and analyzed routing proportions across benchmarks, demonstrating Lookahead’s specialization awareness and effectiveness on hard samples. We also conducted a bad case analysis and discussed future improvement directions.
- **Novelty and Relevance** (`1mHC`): We emphasized Lookahead’s paradigm shift from query-only routing to response-anticipation, with latent representations acting as a causal bridge for optimal model selection. Routing remains critical due to the complementary strengths of specialized models, supported by recent studies and adoption in practical systems and platforms (e.g., GPT-5, Not Diamond).

All of these results and discussions will be incorporated into the revised manuscript to enhance its clarity and completeness.

Overall, our work introduces the first framework leveraging latent output representations for informed model selection. By enabling "foresight" into potential model outputs, Lookahead advances more effective multi-model collaboration, with broad implications for optimizing LLM deployments across diverse tasks.

---

### Decision · Program_Chairs · 2025-09-17

**Decision:**

Accept (poster)

**Comment:**

The paper proposes a new framework for LLM routing that leverages predicted responses rather than relying solely on queries. The efficacy of the approach is well demonstrated through extensive empirical studies. Overall, the reviewers are positive about the work, and I agree that LLM routing is a promising direction, particularly as both open-source and closed-source models continue to proliferate. Therefore, I recommend acceptance.